# MCEVAL: MASSIVELY MULTILINGUAL CODE EVALUATION

**Linzheng Chai**[1][*], **Shukai Liu**[1][*], **Jian Yang**[1][*][†], **Yuwei Yin**[2], **Ke Jin**[1], **Jiaheng Liu**[1],
**Tao Sun**[1], **Ge Zhang**[3], **Changyu Ren**[1], **Hongcheng Guo**[1], **Zekun Wang**[1], **Boyang Wang**[1],
**Xianjie Wu**[1], **Bing Wang**[1], **Tongliang Li**[4], **Liqun Yang**[1], **Sufeng Duan**[5], **Zhoujun Li**[1]
[1]CCSE, Beihang University, [2]University of British Columbia, [3]University of Waterloo
[4]Beijing Information Science and Technology University, [5]Shanghai Jiao Tong University

## ABSTRACT

Code large language models (LLMs) have shown remarkable advances in code understanding, completion, and generation tasks. Programming benchmarks, comprised of a selection of code challenges and corresponding test cases, serve as a standard to evaluate the capability of different LLMs in such tasks. However, most existing benchmarks primarily focus on Python and are still restricted to a limited number of languages, where other languages are translated from the Python samples degrading the data diversity. To further facilitate the research of code LLMs, we propose a massively multilingual code benchmark covering 40 programming languages (MCEVAL) with 16K test samples, which substantially pushes the limits of code LLMs in multilingual scenarios. The benchmark contains challenging code completion, understanding, and generation evaluation tasks with finely curated massively multilingual instruction corpora MCEVAL-INSTRUCT. In addition, we introduce an effective multilingual coder MCODER trained on MCEVAL-INSTRUCT to support multilingual programming language generation. Extensive experimental results on MCEVAL show that there is still a difficult journey between open-source models and closed-source LLMs in numerous languages. The instruction corpora and evaluation benchmark are available at `https://github.com/MCEVAL/McEval`.

## 1 INTRODUCTION

Large language models (LLMs) designed for code, such as Codex (Chen et al., 2021), CodeGen (Nijkamp et al., 2023), Code Llama (Rozière et al., 2023), DeepSeek-Coder (Guo et al., 2024), and CodeQwen (Hui et al., 2024) excel at code understanding, completion, and generation tasks.

Code LLMs with a large number of parameters (e.g. 7B, 13B, or larger) are pre-trained on large-scale code databases with self-supervised autoregressive objectives, followed by instruction tuning (Ouyang et al., 2022) for aligning to human preferences and downstream code-related tasks. Most code benchmarks (Chen et al., 2021; Austin et al., 2021; Athiwaratkun et al., 2023) are introduced to evaluate the performance of code LLMs by assessing their ability to generate executable code based on the problem descriptions. The assessments aim to gauge the capacity of the models to understand and generate code effectively, thereby contributing to facilitating and streamlining the programming process for developers. The execution-based method executes generated code against test cases to measure the

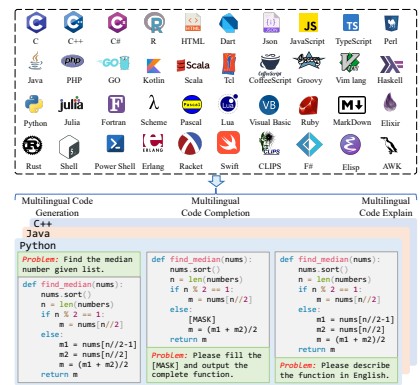

Figure 1: MCEVAL comprised three tasks: code generation, code completion, and code explanation.

---

[*] Equal contribution.
[†] Corresponding Author.

success rate. Due to the difficulty of creating the problem and its corresponding solution (requiring specialized programming staff), the development of evaluation benchmarks is limited within Python, with a few other languages being translated from Python. *Therefore, the community desperately needs a massively multilingual programming benchmark (not from HumanEval or MBPP) comprised of instruction corpora and evaluation set to comprehensively facilitate and evaluate the generation, completion, and understanding capability of LLMs.*

To facilitate the development of code LLMs, we introduce a complete framework that includes the multilingual code instruction corpora, multilingual coder (MCODER), and multilingual code evaluation benchmark. First, we propose MCEVAL, the first massively multilingual code evaluation benchmark (from human handwriting) covering 40 languages (16K samples in total), encompassing multilingual code generation, multilingual code explanation, and multilingual code completion tasks. Then, we create a massively multilingual instruction corpora MCEVAL-INSTRUCT of 40 languages. We initially select and refine high-quality code snippets from various programming languages (PLs) using an LLM. The LLM then generates clear and self-contained instructional content, including problem descriptions and corresponding solutions, based on the refined snippets. To ensure consistency and enhance learning across languages, we introduce cross-lingual code transfer, adapting instructional content to different PLs while increasing sample complexity. Based on open-source models and MCEVAL-INSTRUCT, MCODER is used as a strong baseline to explore the transferability of LLMs among different PLs.

The contributions are summarized as follows: (1) We propose MCEVAL with enough test samples (16K), a true massively multilingual multitask code evaluation benchmark (not from HumanEval or MBPP) covering 40 languages, encompassing multilingual code generation, multilingual code explanation, and multilingual code completion tasks. (2) We introduce MCEVAL-INSTRUCT, the massively multilingual code instruction corpora covering from the multilingual code snippet from 40 languages. Based on MCEVAL-INSTRUCT, an effective multilingual coder MCODER is used as a strong baseline for MCEVAL. (3) We systematically evaluate the understanding and generation capabilities of 20+ models on our created MCEVAL and create a leaderboard to evaluate them on 40 programming languages dynamically. Notably, extensive experiments suggest that comprehensive multilingual multitask evaluation can realistically measure the gap between open-source (e.g. DeepSeekCoder and CodeQwen1.5) and closed-source models (e.g. GPT-3.5 and GPT-4).

## 2 MULTILINGUAL CODE EVALUATION: MCEVAL

### 2.1 DATASET STATISTICS

The created MCEVAL is comprised of three key code-related tasks covering 40 programming languages, including multilingual code generation, multilingual code explanation, and multilingual code completion tasks. The multilingual code generation and explanation tasks separately contain 2K samples, where each language has nearly 50 samples. The code completion task can be decomposed into *multi-line completion* (3K samples), *single-line completion* (3K samples), *span completion* (4K samples), and *span completion (light)* (2K samples) (Bavarian et al., 2022).

In Table 1, we display the number of questions, test cases, and difficulty levels corresponding to the three tasks in MCEVAL and the number of questions in the four sub-tasks of the completion task. Moreover, we counted the token length of the prompt and solutions. (The tokens are calculated based on the Llama-3 tokenizer.) Among these tasks, the *span completion (light)* task is similar in form to the *span completion* task. However, in the *span completion (light)* task, each problem is paired with all the corresponding code, making it a balanced version of the *span completion* task (fewer samples for fast inference and the same test size of

Table 1: MCEVAL dataset statistics.

| Statistics | Value |
| --- | --- |
| **Questions** | |
| Code Generation | $2,007$ |
| Code Explanation | $2,007$ |
| Code Completion | $12,017$ |
| *- Single-Line* | $2,998$ |
| *- Multi-Line* | $2,998$ |
| *- Span* | $4,014$ |
| *- Span(light)* | $2,007$ |
| Total Test Cases | $10,086$ |
| **Difficulty Level** | |
| - Easy | $1,221$ |
| - Medium | $401$ |
| - Hard | $385$ |
| **Length** | |
| Prompt | |
| *- maximum length* | 793 tokens |
| *- minimum length* | 16 tokens |
| *- avg length* | 173.8 tokens |
| Solution(Output) | |
| *- maximum length* | 666 tokens |
| *- minimum length* | 4 tokens |
| *- avg length* | 120.9 tokens |

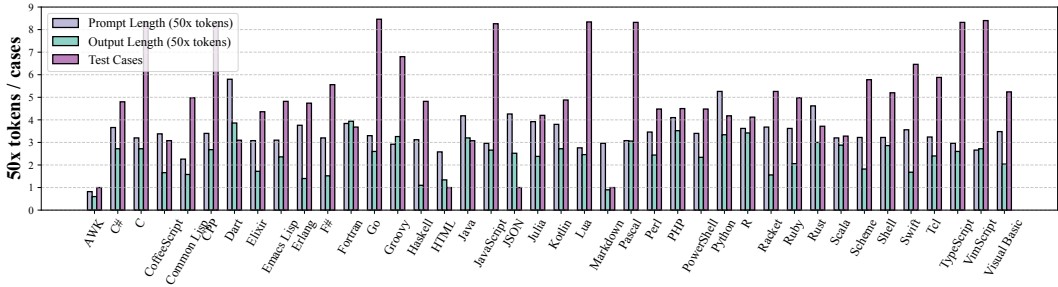

Figure 2: Data statistics of the MCEVAL benchmark involving 40 programming languages.

each programming language). The results of *span completion (light)* can better reflect the differences in model performance across different languages.

Figure 2 plots the length of the prompt, solution(output), and the number of test cases of each programming language. In Table 2, We compared MCEVAL with other multilingual benchmarks. It is noteworthy that our benchmark provides a significant supplement to current benchmarks in terms of both the variety of programming languages and the number of questions.

Table 2: Comparison between MCEVAL and other multilingual code benchmarks. $^\diamond$The number of each of the three tasks (Generation, Explanation, and Completion).

| Benchmark | Multi-Task | #Languages | Data source | #Questions |
|---|---|---|---|---|
| MuliPL-E (Cassano et al., 2023) | ✗ | 18 | Translate | ~3,000 |
| MBXP Athiwaratkun et al. (2023) | ✓ | 10 | Translate | 12,425 |
| HumanEval-X Zheng et al. (2023b) | ✓ | 5 | Hand-Written | 820 |
| HumanEval-XL Peng et al. (2024) | ✗ | 12 | Hand-Written | 22,080 |
| MCEVAL | ✓ | 40 | Hand-Written | 16,031 (2007/2007/12017)$^\diamond$ |

## 2.2 HUMAN ANNOTATION & QUALITY CONTROL

To create the massively multilingual code evaluation benchmark MCEVAL, the annotation of multilingual code samples is conducted utilizing a comprehensive and systematic human annotation procedure, underpinned by rigorously defined guidelines to ensure accuracy and consistency. Initially, 10 software developers in computer science are recruited as multilingual programming annotators with proven proficiency in the respective programming languages. Following a detailed training session on the annotation protocol, which emphasizes the importance of context, syntactical correctness, and semantic fidelity across languages, annotators are tasked with creating problem definitions and the corresponding solution. The annotators should follow: (1) Provide a clear and self-contained problem definition, answer the question with any tools, and design the test cases to evaluate the correctness of the code. (2) Classify them into multiple difficulties (Easy/Middle/Hard), based on algorithmic complexity and functionality. Each sample is independently annotated by at least two annotators to minimize subjective bias and errors. Discrepancies between annotators are resolved through consensus or adjudication by a senior annotator. Finally, three volunteers are employed to evaluate the correctness of the benchmark (> 90% accuracy) and correct the errors. (See Appendix A.2 for more details).

## 2.3 EVALUATION TASKS

**Multilingual Code Generation.** Given the $k$-th programming language $L_k \in \{L_i\}_{i=1}^{K}$, where $K = 40$ is the number of programming languages, we provide the problem description $q^{L_k}$ and examples test cases $e^{L_k}$ as the input for code LLMs $\mathcal{M}$ to generate the corresponding code $a^{L_k}$. We obtain the sampled code result from the code generation distribution $P(a^{L_k}|q^{L_k}, e^{L_k}; \mathcal{M})$ from code LLM $\mathcal{M}$, and then feed the test cases into the generated code, where the generated outputs by code should equal the expected outputs. The process can be described as:

$$r^{L_k} = \mathbb{I}(P(a^{L_k}|q^{L_k}, e^{L_k}; \mathcal{M}); u^{L_k}) \tag{1}$$

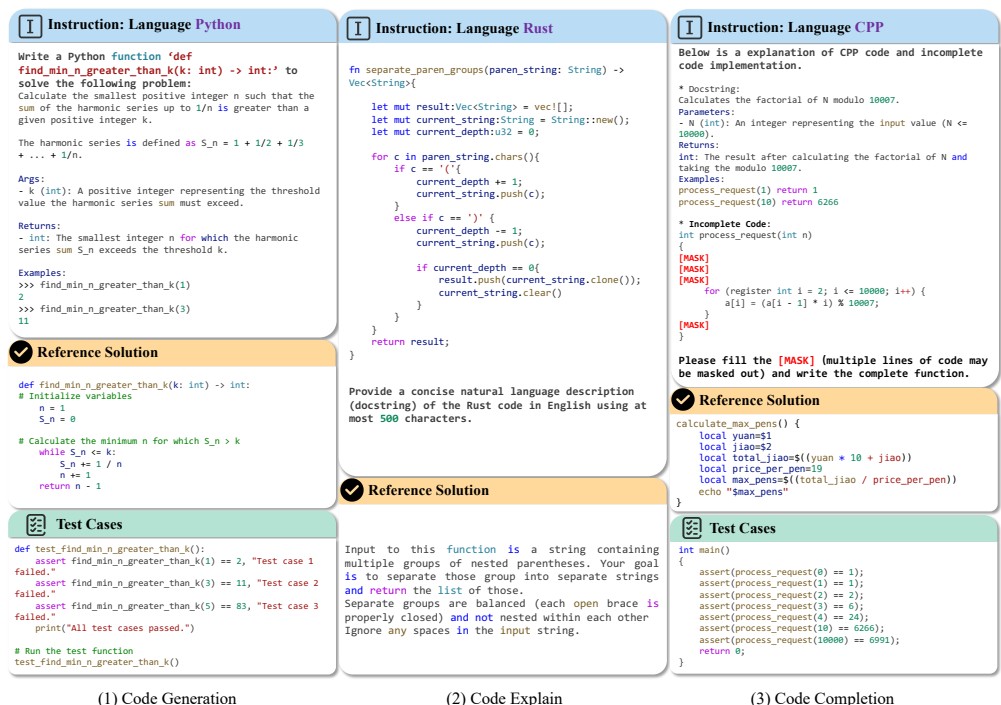

Figure 3: Examples of multilingual code generation, explanation, and completion.

where $\mathbb{I}(\cdot)$ is the indicator function by executing the generated code with the given test cases $u^{L_k}$. when the generated code $a^{L_k}$ passes all test cases, the evaluation result $r = 1$, else $r = 0$.

**Multilingual Code Explanation.** To evaluate the understanding capability of code LLMs, we adopt two-pass generation (Code-to-Natural-Language and Natural-Language-to-Code), since the text-similar metrics (e.g. BLEU (Papineni et al., 2002) ) are hindered by the $n$-gram text matching and can not produce an accurate score. We first prompt the code LLMs to generate the natural language description $t^{L_k}$ based on the code $a^{L_k}$ and then we force the model to restore the original code based on $t^{L_k}$. The sampled code from $P(a^{L_k}|t^{L_k}; \mathcal{M})$ is used to evaluate the understanding capability as:

$$r = \mathbb{I}(P(t^{L_k}|a^{L_k}; \mathcal{M})P(a^{L_k}|t^{L_k}; \mathcal{M}); u^{L_k}) \qquad (2)$$

where $\mathbb{I}(\cdot)$ is used to check the correctness of the generated code by running the code with test cases.

**Multilingual Code Completion.** Another important scenario is code completion, where the code LLM produces the middle code $a_m^{L_k}$ based on the prefix code $a_p^{L_k}$ and suffix code snippet $a_s^{L_k}$. Hence, we concatenate $a_p^{L_k}$, $a_m^{L_k}$, and $a_s^{L_k}$ as the complete code for evaluation as:

$$r = \mathbb{I}(P(a_m^{L_k}|a_p^{L_k}, a_s^{L_k}; \mathcal{M}); u^{L_k}) \qquad (3)$$

where $a_p^{L_k}$, $a_q^{L_k}$, and $a_m^{L_k}$ are concatenated as the complete code to be executed with test cases $u^{L_k}$.

## 3 MCODER

### 3.1 MCEVAL-INSTRUCT

**Collection from Code Snippet.** For a programming language $L_k$ ($L_k \in \{L_i\}_{i=1}^K$) and $K$ is the number of programming languages), consider an existing code snippet $c \in D_c^{L_k}$, we prompt the LLM to select the high-quality code and refine the code to a self-contained code snippet by using the prompt "{Code Snippet}\nDetermine its educational value for a student whose goal is to learn basic coding concepts.\n\nIf the answer is 'YES'. Please refine the code with clear variable definitions, comments, and docstring.". Then, we can obtain the multilingual refined code snippets. (More details can be found in Appendix A.3)

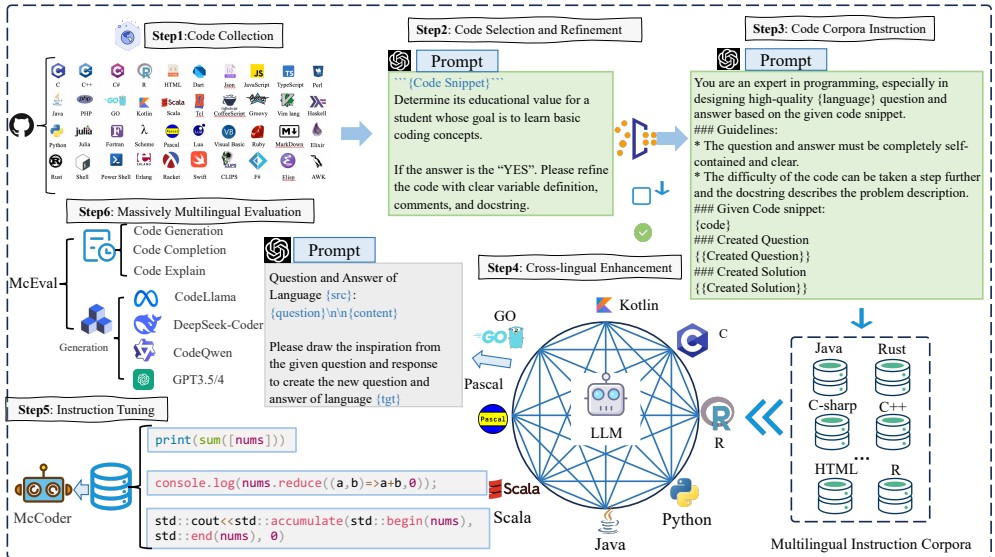

Figure 4: The framework of MCODER. We first create MCEVAL-INSTRUCT covering 40 languages from code snippets to fine-tune MCODER. 20+ existing LLMs and MCODER are then evaluated on MCEVAL comprised of multilingual code generation, explanation, and completion.

**Instruction Corpora Generation.** To construct a comprehensive massively multilingual code instruction corpora $\{D^{L_i}\}_{i=1}^K$, we prompt the LLMs (gpt-4-1106-preview) to create a problem description $q^{L_k}$ and the corresponding solution $a^{L_k}$ by drawing inspiration from the refined code snippet $c^{L_k}$. We use LLM to generate instruction dataset by using the prompt "You are an expert in programming, especially in designing high-quality language question and answer based on the given code snippet.\n\n ### Guidelines: * The question and answer must be completely self-contained and clear.*\n The difficulty of the code can be taken a step further and the docstring describes the problem description.\n ### Given Code snippet: code\n ### Created Question Created Question\n ### Created Solution\n Created Solution" in Figure 4.

**Cross-lingual Code Transfer.** Since the created instruction samples of different programming languages focus on different aspects of coding, we adopt the cross-lingual code transfer to minimize the gap among multiple languages. Given the instruction dataset $D^{L_i}$ of language $L_i$, we randomly sample a pair $(q^{L_i}, a^{L_i})$ and force the LLM to modify them to another language $L_j$ with a more complex sample $(q^{L_i \to L_j}, a^{L_i \to L_j})$. In this way, we can get the derived instruction corpora $\{D^{L_i \to L_j}\}(i \neq j \land 1 \leq i,j \leq K)$. Finally, we combine $\{D^{L_k}\}_{k=1}^K$ and $\{D^{L_i \to L_j}\}(i \neq j \land 1 \leq i,j \leq K)$ as the multilingual instruction corpora MCEVAL-INSTRUCT $\{D_{mc}^{L_k}\}_{k=1}^K$ covering 40 programming languages.

## 3.2 MULTILINGUAL CODE INSTRUCTION TUNING

The training objective $\mathcal{L}_{all}$ of the multilingual instruction fine-tuning can be described as:

$$\mathcal{L}_{all} = -\sum_{k=1}^K \mathbb{E}_{q^{L_k}, a^{L_k} \sim \{D^{L_k}\}_{k=1}^K} \left[ \log P(a^{L_k} | q^{L_k}; \mathcal{M}) \right] \quad (4)$$

where $q^{L_k}$ and $a^{L_k}$ are the code-related question and answer from the dataset $D^{L_k}$ of language $L_k$, respectively. $K$ is the number of programming languages.

## 4 EXPERIMENTS

## 4.1 EXPERIMENT SETUP

**Code LLMs.** We evaluate 30 + models with sizes ranging from 7B to 236B parameters, including general/code LLMs, open/closed-source models, and base/instruction models. For general models, we evaluate GPT series (Brown et al., 2020; OpenAI, 2023), Qwen1.5 (Bai et al., 2023), Llama3 (AI,

Table 3: Pass@1 (%) scores of different code LLMs for multilingual code generation tasks on MCEVAL. "Avg$_{all}$" represents the average scores of all code languages.

| Method | Size | AWK | C | C++ | C# | Clisp | Coffee | Dart | Elisp | Elixir | Erlang | Fortran | F# | Go | Groovy | Haskell | Html | Java | JS | Json | Julia |
|---|---|---|---|---|---|---|---|---|---|---|---|---|---|---|---|---|---|---|---|---|---|
| GPT-4o (240513) | 🔒 | 54.0 | 60.0 | 58.0 | 72.0 | 60.0 | 82.0 | 54.9 | 64.0 | 66.0 | 44.0 | 66.0 | 78.0 | 62.0 | 80.0 | 90.0 | 32.0 | 81.1 | 62.0 | 74.0 | 72.0 |
| GPT-4 Turbo (231106) | 🔒 | 70.0 | 60.0 | 64.0 | 80.0 | 64.0 | 72.0 | 45.1 | 62.0 | 56.0 | 38.0 | 54.0 | 74.0 | 56.0 | 82.0 | 78.0 | 30.0 | 83.0 | 60.0 | 72.0 | 70.0 |
| GPT-3.5 Turbo (240125) | 🔒 | 14.0 | 54.0 | 58.0 | 68.0 | 54.0 | 76.0 | 41.2 | 26.0 | 30.0 | 46.0 | 40.0 | 68.0 | 54.0 | 86.0 | 50.0 | 18.0 | 71.7 | 60.0 | 76.0 | 54.0 |
| Yi-Large-Turbo | 🔒 | 52.0 | 44.0 | 50.0 | 54.0 | 54.0 | 52.0 | 31.4 | 30.0 | 46.0 | 30.0 | 58.0 | 52.0 | 46.0 | 80.0 | 58.0 | 22.0 | 41.5 | 52.0 | 78.0 | 56.0 |
| DeepSeekCoder-V2-Instruct | 236B | 62.0 | 60.0 | 66.0 | 72.0 | 74.0 | 80.0 | 43.1 | 52.0 | 62.0 | 32.0 | 80.0 | 76.0 | 56.0 | 84.0 | 72.0 | 30.0 | 77.4 | 64.0 | 74.0 | 70.0 |
| Qwen1.5-Chat | 72B | 50.0 | 48.0 | 46.0 | 56.0 | 46.0 | 44.0 | 19.6 | 14.0 | 18.0 | 24.0 | 34.0 | 32.0 | 50.0 | 42.0 | 32.0 | 22.0 | 39.6 | 46.0 | 74.0 | 32.0 |
| CodeLlama-Instruct | 34B | 38.0 | 32.0 | 32.0 | 40.0 | 42.0 | 34.0 | 7.8 | 16.0 | 28.0 | 32.0 | 24.0 | 18.0 | 34.0 | 20.0 | 32.0 | 14.0 | 26.4 | 42.0 | 68.0 | 28.0 |
| WizardCoder-Python | 34B | 36.0 | 42.0 | 46.0 | 52.0 | 42.0 | 46.0 | 13.7 | 14.0 | 38.0 | 38.0 | 26.0 | 26.0 | 50.0 | 54.0 | 40.0 | 26.0 | 43.4 | 52.0 | 70.0 | 52.0 |
| DeepSeekCoder-Instruct | 33B | 50.0 | 58.0 | 66.0 | 70.0 | 60.0 | 86.0 | 25.5 | 48.0 | 50.0 | 40.0 | 66.0 | 48.0 | 54.0 | 78.0 | 56.0 | 30.0 | 73.6 | 62.0 | 42.0 | 56.0 |
| Codestral-v0.1 | 22B | 54.0 | 48.0 | 56.0 | 66.0 | 60.0 | 62.0 | 43.1 | 28.0 | 34.0 | 24.0 | 56.0 | 58.0 | 58.0 | 76.0 | 52.0 | 34.0 | 73.6 | 58.0 | 72.0 | 52.0 |
| DeepSeekCoder-V2-Lite-Instruct | 16B | 58.0 | 58.0 | 64.0 | 74.0 | 56.0 | 76.0 | 35.3 | 30.0 | 40.0 | 26.0 | 68.0 | 56.0 | 56.0 | 60.0 | 60.0 | 26.0 | 66.0 | 64.0 | 68.0 | 60.0 |
| OCTOCODER | 16B | 28.0 | 28.0 | 28.0 | 38.0 | 40.0 | 18.0 | 5.9 | 14.0 | 28.0 | 16.0 | 22.0 | 4.0 | 34.0 | 30.0 | 20.0 | 8.0 | 34.0 | 30.0 | 58.0 | 20.0 |
| WizardCoder-V1.0 | 15B | 18.0 | 38.0 | 28.0 | 36.0 | 36.0 | 50.0 | 17.6 | 0.0 | 24.0 | 24.0 | 28.0 | 30.0 | 38.0 | 62.0 | 40.0 | 6.0 | 30.2 | 52.0 | 14.0 | 38.0 |
| Granite-34B-code-instruct-8K | 34B | 48.0 | 38.0 | 54.0 | 50.0 | 60.0 | 64.0 | 19.6 | 16.0 | 36.0 | 36.0 | 50.0 | 40.0 | 52.0 | 64.0 | 44.0 | 28.0 | 37.7 | 54.0 | 66.0 | 48.0 |
| Granite-20B-code-instruct-8K | 20B | 38.0 | 42.0 | 40.0 | 56.0 | 40.0 | 56.0 | 23.5 | 18.0 | 40.0 | 16.0 | 32.0 | 42.0 | 46.0 | 48.0 | 40.0 | 20.0 | 35.8 | 50.0 | 74.0 | 48.0 |
| Granite-8B-code-instruct-4K | 8B | 36.0 | 28.0 | 44.0 | 48.0 | 50.0 | 56.0 | 11.8 | 20.0 | 36.0 | 22.0 | 32.0 | 38.0 | 46.0 | 48.0 | 40.0 | 26.0 | 35.8 | 48.0 | 64.0 | 48.0 |
| Granite-3B-code-instruct-128K | 3B | 16.0 | 14.0 | 42.0 | 38.0 | 40.0 | 30.0 | 9.8 | 12.0 | 36.0 | 16.0 | 26.0 | 26.0 | 30.0 | 32.0 | 34.0 | 22.0 | 26.4 | 50.0 | 62.0 | 28.0 |
| Phi-3-medium-4k-instruct | 14B | 52.0 | 46.0 | 40.0 | 52.0 | 34.0 | 42.0 | 13.7 | 14.0 | 16.0 | 8.0 | 30.0 | 32.0 | 42.0 | 42.0 | 20.0 | 39.6 | 52.0 | 68.0 | 48.0 | |
| CodeLlama-Instruct | 13B | 36.0 | 38.0 | 38.0 | 40.0 | 46.0 | 30.0 | 7.8 | 16.0 | 32.0 | 32.0 | 16.0 | 26.0 | 34.0 | 22.0 | 38.0 | 18.0 | 22.6 | 34.0 | 56.0 | 32.0 |
| Llama-3-Instruct | 8B | 32.0 | 46.0 | 50.0 | 54.0 | 38.0 | 48.0 | 15.7 | 14.0 | 32.0 | 30.0 | 12.0 | 26.0 | 48.0 | 52.0 | 38.0 | 16.0 | 45.3 | 54.0 | 70.0 | 40.0 |
| CodeQwen-2.5-Chat | 7B | 58.0 | 58.0 | 58.0 | 74.0 | 56.0 | 86.0 | 37.3 | 54.0 | 56.0 | 38.0 | 56.0 | 62.0 | 58.0 | 84.0 | 66.0 | 34.0 | 69.8 | 60.0 | 76.0 | 64.0 |
| Codegemma-it | 7B | 26.0 | 40.0 | 42.0 | 48.0 | 20.0 | 18.0 | 23.5 | 10.0 | 4.0 | 4.0 | 8.0 | 22.0 | 46.0 | 58.0 | 32.0 | 24.0 | 56.6 | 48.0 | 70.0 | 38.0 |
| CodeLlama-Instruct | 7B | 32.0 | 34.0 | 26.0 | 40.0 | 42.0 | 32.0 | 5.9 | 14.0 | 22.0 | 20.0 | 20.0 | 14.0 | 32.0 | 22.0 | 32.0 | 14.0 | 18.9 | 28.0 | 64.0 | 24.0 |
| Codeshell-chat | 7B | 24.0 | 30.0 | 36.0 | 26.0 | 20.0 | 38.0 | 5.9 | 4.0 | 14.0 | 6.0 | 8.0 | 8.0 | 28.0 | 30.0 | 22.0 | 24.0 | 22.6 | 42.0 | 66.0 | 24.0 |
| DeepSeekCoder-1.5-Instruct | 7B | 40.0 | 54.0 | 56.0 | 60.0 | 56.0 | 80.0 | 23.5 | 24.0 | 40.0 | 40.0 | 40.0 | 46.0 | 52.0 | 80.0 | 26.0 | 24.0 | 60.4 | 56.0 | 66.0 | 42.0 |
| Magicoder-S-DS | 7B | 44.0 | 50.0 | 50.0 | 60.0 | 58.0 | 72.0 | 19.6 | 32.0 | 34.0 | 62.0 | 62.0 | 54.0 | 50.0 | 80.0 | 54.0 | 16.0 | 66.0 | 60.0 | 56.0 | 40.0 |
| Nxcode-CQ-orpo | 7B | 38.0 | 52.0 | 58.0 | 50.0 | 46.0 | 62.0 | 23.5 | 22.0 | 38.0 | 36.0 | 52.0 | 46.0 | 52.0 | 72.0 | 46.0 | 28.0 | 56.6 | 50.0 | 64.0 | 62.0 |
| OpenCodeInterpreter-DS | 7B | 38.0 | 54.0 | 52.0 | 68.0 | 44.0 | 78.0 | 17.6 | 30.0 | 42.0 | 48.0 | 52.0 | 54.0 | 48.0 | 72.0 | 40.0 | 28.0 | 66.0 | 46.0 | 72.0 | 34.0 |
| CodeQwen-1.5-Chat | 7B | 40.0 | 52.0 | 56.0 | 62.0 | 48.0 | 62.0 | 29.4 | 22.0 | 38.0 | 38.0 | 50.0 | 44.0 | 50.0 | 70.0 | 44.0 | 30.0 | 58.5 | 54.0 | 64.0 | 62.0 |
| MCODER (Our Method) | 7B | 40.0 | 44.0 | 52.0 | 62.0 | 46.0 | 66.0 | 21.6 | 30.0 | 44.0 | 58.0 | 56.0 | 44.0 | 48.0 | 70.0 | 32.0 | 34.0 | 54.7 | 54.0 | 66.0 | 56.0 |

| Method | Kotlin | Lua | MD | Pascal | Perl | PHP | Power | Python | R | Racket | Ruby | Rust | Scala | Scheme | Shell | Swift | Tcl | TS | VB | VimL | Avg$_{all}$ |
|---|---|---|---|---|---|---|---|---|---|---|---|---|---|---|---|---|---|---|---|---|---|
| GPT-4o (240513) | 84.0 | 60.0 | 32.0 | 52.0 | 64.0 | 64.0 | 72.0 | 76.0 | 66.0 | 66.0 | 64.0 | 83.0 | 32.0 | 76.0 | 76.0 | 84.0 | 68.0 | 56.0 | 78.0 | 40.0 | 65.2 |
| GPT-4 Turbo (231106) | 80.0 | 56.0 | 38.0 | 46.0 | 64.0 | 68.0 | 76.0 | 78.0 | 58.0 | 62.0 | 66.0 | 71.7 | 66.0 | 58.0 | 72.0 | 82.0 | 56.0 | 60.0 | 68.0 | 38.0 | 63.4 |
| GPT-3.5 Turbo (240125) | 62.0 | 58.0 | 24.0 | 40.0 | 56.0 | 58.0 | 68.0 | 60.0 | 56.0 | 40.0 | 56.0 | 52.8 | 68.0 | 46.0 | 60.0 | 58.0 | 44.0 | 54.0 | 74.0 | 24.0 | 52.6 |
| Yi-Large-Turbo | 36.0 | 48.0 | 24.0 | 46.0 | 44.0 | 46.0 | 64.0 | 44.0 | 50.0 | 34.0 | 64.0 | 0.0 | 48.0 | 50.0 | 48.0 | 64.0 | 52.0 | 50.0 | 40.0 | 30.0 | 46.6 |
| DeepSeekCoder-V2-Instruct | 80.0 | 58.0 | 36.0 | 54.0 | 54.0 | 68.0 | 78.0 | 64.0 | 64.0 | 66.0 | 64.0 | 84.9 | 76.0 | 74.0 | 70.0 | 74.0 | 68.0 | 50.0 | 72.0 | 40.0 | 64.6 |
| Qwen1.5-Chat | 20.0 | 50.0 | 22.0 | 36.0 | 40.0 | 36.0 | 40.0 | 32.0 | 22.0 | 26.0 | 44.0 | 34.0 | 30.0 | 30.0 | 34.0 | 40.0 | 26.0 | 44.0 | 30.0 | 28.0 | 35.8 |
| CodeLlama-Instruct | 24.0 | 30.0 | 10.0 | 28.0 | 40.0 | 24.0 | 38.0 | 32.0 | 18.0 | 26.0 | 30.0 | 26.4 | 28.0 | 20.0 | 20.0 | 42.0 | 24.0 | 38.0 | 36.0 | 22.0 | 29.1 |
| WizardCoder-Python | 36.0 | 46.0 | 12.0 | 36.0 | 36.0 | 38.0 | 44.0 | 40.0 | 20.0 | 22.0 | 46.0 | 35.8 | 48.0 | 22.0 | 2.0 | 48.0 | 30.0 | 28.0 | 40.0 | 24.0 | 36.5 |
| DeepSeekCoder-Instruct | 66.0 | 58.0 | 32.0 | 54.0 | 32.0 | 48.0 | 60.0 | 56.0 | 48.0 | 46.0 | 66.0 | 66.0 | 62.0 | 60.0 | 58.0 | 60.0 | 48.0 | 72.0 | 44.0 | 30.0 | 54.3 |
| Codestral-v0.1 | 64.0 | 56.0 | 16.0 | 6.0 | 54.0 | 56.0 | 64.0 | 56.0 | 48.0 | 46.0 | 71.7 | 48.0 | 32.0 | 48.0 | 72.0 | 44.0 | 52.0 | 62.0 | 28.0 | 50.5 | |
| DeepSeekCoder-V2-Lite-Instruct | 64.0 | 56.0 | 28.0 | 48.0 | 56.0 | 44.0 | 68.0 | 64.0 | 50.0 | 52.0 | 56.0 | 71.7 | 60.0 | 54.0 | 52.0 | 76.0 | 38.0 | 60.0 | 58.0 | 24.0 | 54.7 |
| OCTOCODER | 14.0 | 32.0 | 10.0 | 6.0 | 26.0 | 24.0 | 38.0 | 30.0 | 6.0 | 24.0 | 0.0 | 32.1 | 4.0 | 22.0 | 26.0 | 30.0 | 24.0 | 38.0 | 28.0 | 14.0 | 23.3 |
| WizardCoder-V1.0 | 26.0 | 40.0 | 0.0 | 12.0 | 32.0 | 32.0 | 48.0 | 30.0 | 20.0 | 10.0 | 40.0 | 24.5 | 40.0 | 10.0 | 0.0 | 30.0 | 20.0 | 42.0 | 36.0 | 18.0 | 28.0 |
| Granite-34B-code-instruct-8K | 44.0 | 50.0 | 22.0 | 40.0 | 34.0 | 40.0 | 56.0 | 44.0 | 32.0 | 36.0 | 38.0 | 45.3 | 34.0 | 36.0 | 36.0 | 52.0 | 32.0 | 50.0 | 44.0 | 16.0 | 42.2 |
| Granite-20B-code-instruct-8K | 38.0 | 48.0 | 22.0 | 32.0 | 34.0 | 40.0 | 48.0 | 44.0 | 28.0 | 32.0 | 40.0 | 39.6 | 36.0 | 40.0 | 30.0 | 46.0 | 28.0 | 46.0 | 38.0 | 16.0 | 38.3 |
| Granite-8B-code-instruct-4K | 32.0 | 48.0 | 20.0 | 32.0 | 38.0 | 34.0 | 42.0 | 36.0 | 22.0 | 24.0 | 36.0 | 43.4 | 40.0 | 36.0 | 24.0 | 34.0 | 16.0 | 42.0 | 42.0 | 12.0 | 36.5 |
| Granite-3B-code-instruct-128K | 20.0 | 32.0 | 8.0 | 32.0 | 24.0 | 28.0 | 34.0 | 28.0 | 12.0 | 24.0 | 42.0 | 24.5 | 30.0 | 22.0 | 16.0 | 36.0 | 14.0 | 42.0 | 34.0 | 16.0 | 28.0 |
| Phi-3-medium-4k-instruct | 30.0 | 48.0 | 18.0 | 28.0 | 36.0 | 42.0 | 50.0 | 48.0 | 38.0 | 30.0 | 26.0 | 26.4 | 18.0 | 30.0 | 26.0 | 50.0 | 22.0 | 52.0 | 44.0 | 12.0 | 35.2 |
| CodeLlama-Instruct | 20.0 | 30.0 | 10.0 | 8.0 | 28.0 | 32.0 | 34.0 | 30.0 | 12.0 | 20.0 | 28.0 | 24.5 | 24.0 | 26.0 | 20.0 | 40.0 | 18.0 | 40.0 | 38.0 | 10.0 | 27.7 |
| Llama-3-Instruct | 34.0 | 40.0 | 14.0 | 32.0 | 36.0 | 38.0 | 40.0 | 42.0 | 30.0 | 22.0 | 34.0 | 41.5 | 38.0 | 32.0 | 26.0 | 48.0 | 24.0 | 50.0 | 42.0 | 16.0 | 36.0 |
| CodeQwen-2.5-Chat | 72.0 | 58.0 | 32.0 | 50.0 | 56.0 | 64.0 | 72.0 | 62.0 | 66.0 | 58.0 | 60.0 | 69.8 | 60.0 | 64.0 | 66.0 | 82.0 | 50.0 | 56.0 | 68.0 | 40.0 | 60.3 |
| Codegemma-it | 36.0 | 48.0 | 8.0 | 14.0 | 40.0 | 36.0 | 42.0 | 24.0 | 16.0 | 18.0 | 40.0 | 39.6 | 40.0 | 20.0 | 12.0 | 54.0 | 10.0 | 38.0 | 38.0 | 16.0 | 30.7 |
| CodeLlama-Instruct | 16.0 | 30.0 | 14.0 | 6.0 | 28.0 | 12.0 | 34.0 | 32.0 | 14.0 | 24.0 | 28.0 | 30.2 | 18.0 | 8.0 | 16.0 | 32.0 | 22.0 | 12.0 | 34.0 | 14.0 | 24.6 |
| Codeshell-chat | 28.0 | 34.0 | 14.0 | 10.0 | 22.0 | 28.0 | 32.0 | 30.0 | 16.0 | 14.0 | 34.0 | 30.2 | 18.0 | 8.0 | 12.0 | 18.0 | 12.0 | 40.0 | 30.0 | 20.0 | 23.0 |
| DeepSeekCoder-1.5-Instruct | 38.0 | 48.0 | 30.0 | 38.0 | 42.0 | 54.0 | 64.0 | 44.0 | 32.0 | 44.0 | 54.0 | 45.3 | 50.0 | 40.0 | 42.0 | 42.0 | 40.0 | 50.0 | 58.0 | 20.0 | 46.0 |
| Magicoder-S-DS | 42.0 | 48.0 | 24.0 | 48.0 | 50.0 | 44.0 | 56.0 | 48.0 | 44.0 | 42.0 | 54.0 | 45.3 | 54.0 | 44.0 | 54.0 | 56.0 | 46.0 | 50.0 | 68.0 | 16.0 | 48.8 |
| Nxcode-CQ-orpo | 42.0 | 46.0 | 18.0 | 42.0 | 38.0 | 40.0 | 44.0 | 44.0 | 36.0 | 42.0 | 46.0 | 47.2 | 44.0 | 40.0 | 42.0 | 66.0 | 30.0 | 48.0 | 60.0 | 18.0 | 44.7 |
| OpenCodeInterpreter-DS | 48.0 | 50.0 | 26.0 | 46.0 | 50.0 | 30.0 | 54.0 | 56.0 | 42.0 | 34.0 | 40.0 | 50.9 | 38.0 | 44.0 | 36.0 | 46.0 | 36.0 | 50.0 | 60.0 | 18.0 | 46.0 |
| CodeQwen-1.5-Chat | 44.0 | 48.0 | 18.0 | 46.0 | 38.0 | 44.0 | 42.0 | 44.0 | 38.0 | 40.0 | 46.0 | 45.3 | 48.0 | 38.0 | 42.0 | 66.0 | 30.0 | 54.0 | 56.0 | 18.0 | 45.5 |
| MCODER (Our Method) | 48.0 | 52.0 | 30.0 | 42.0 | 36.0 | 32.0 | 54.0 | 44.0 | 40.0 | 36.0 | 48.0 | 52.8 | 58.0 | 44.0 | 46.0 | 64.0 | 38.0 | 52.0 | 58.0 | 20.0 | 46.7 |

2024), Phi-3 (Abdin et al., 2024), and Yi (Young et al., 2024). For code models, we test Code-Qwen (Hui et al., 2024), DeepSeekCoder (Guo et al., 2024), CodeLlama (Rozière et al., 2023), OCTOCODER (Muennighoff et al., 2023), CodeShell (Xie et al., 2024), MagiCoder (Wei et al., 2023), WizardCoder (Luo et al., 2023), Codegemma (Gemma Team, 2024) and Granite (Mishra et al., 2024). Furthermore, we further fine-tune MCODER based on CodeQwen1.5 and DeepSeekCoder to explore the language transfer capabilities of code LLMs.

**Evaluation Metrics.** We assess the models by executing the code and evaluating it using the Pass@1 metric. Pass@1 is a widely recognized measure in machine learning, particularly for code generation, as it gauges the model's accuracy in producing correct solutions on the first attempt.

**Instruction Corpora.** The resulting dataset, MCEVAL-INSTRUCT (110K samples), is comprised of created question-answer pairs and open-source collection (Wei et al., 2023). We apply data decontamination before training our MCODER. Following Li et al. (2023); Wei et al. (2023), we adopt the N-gram exact match decontamination method with MCEVAL, HumanEval(Chen et al., 2021), MultiPL-E(Cassano et al., 2023), MBPP(Austin et al., 2021). For supervised fine-tuning (SFT), we utilize CodeQwen-1.5 as the foundational code LLMs. Specifically, we select all Python data from MCEVAL-INSTRUCT, comprising 50K training samples, for MCODER-Python training.

**Optimization & Evaluation.** Our MCODER based on CodeQwen1.5 are trained for 2 epochs with a cosine scheduler, starting at a learning rate of 2e-5 (3% warmup steps). We use AdamW (Loshchilov & Hutter, 2017) as the optimizer and a batch size of 512 (max length 4096). We adopt the greedy Pass@1 (%) metric (Kulal et al., 2019; Chen et al., 2021) for evaluations. For closed-source LLMs, the answers are generated by the official API. For code explanation, we prompt the LLM to describe the code and then restore the descriptions to the original code. (Details can be found in Appendix A.6).

## 4.2 MAIN RESULTS

**Multilingual Code Generation.** Table 3 shows the Pass@1 results of various models on MCEVAL for multilingual code generation task. The results reveal a significant disparity between closed-source

Table 4: Pass@1 (%) scores of different models for multilingual code completion tasks on MCEVAL. "Avg$_{all}$" represents the average scores of all code languages.

**Single-line Completion**

| Method | Size | AWK | C | C++ | C# | Clisp | Coffee | Dart | Elisp | Elixir | Erlang | Fortran | F# | Go | Groovy | Haskell | Html | Java | JS | Json | Julia |
|---|---|---|---|---|---|---|---|---|---|---|---|---|---|---|---|---|---|---|---|---|---|
| GPT-4 Turbo (231106) | 🔒 | 92.9 | 74.4 | 75.6 | 89.3 | 91.1 | 97.5 | 76.5 | 84.2 | 82.4 | 54.2 | 79.6 | 69.6 | 81.7 | 92.6 | 76.2 | 57.1 | 93.9 | 80.0 | 93.1 | 93.3 |
| GPT-3.5 Turbo (240125) | 🔒 | 14.3 | 32.9 | 20.7 | 41.7 | 53.6 | 68.8 | 28.6 | 44.7 | 19.1 | 6.3 | 14.3 | 37.5 | 67.1 | 76.6 | 11.9 | 21.4 | 40.8 | 35.0 | 4.2 | 76.7 |
| DeepSeekCoder-Instruct | 33B | 50.0 | 61.0 | 72.0 | 79.8 | 62.5 | 78.8 | 70.4 | 63.2 | 66.2 | 45.8 | 68.4 | 60.7 | 67.1 | 83.0 | 38.1 | 52.4 | 82.7 | 65.0 | 77.8 | 83.3 |
| OCTOCODER | 16B | 42.9 | 47.6 | 52.4 | 82.1 | 35.7 | 56.3 | 60.2 | 34.2 | 8.8 | 0.0 | 0.0 | 48.2 | 58.5 | 73.4 | 0.0 | 2.4 | 81.6 | 56.3 | 6.9 | 0.0 |
| StarCoder2-instruct-V0.1 | 15B | 28.6 | 74.4 | 81.7 | 86.9 | 71.4 | 7.5 | 82.7 | 68.4 | 75.0 | 62.5 | 76.5 | 64.3 | 82.9 | 88.3 | 47.6 | 0.0 | 91.8 | 83.8 | 16.7 | 83.3 |
| WizardCoder-V1.0 | 15B | 21.4 | 37.8 | 36.6 | 38.1 | 3.6 | 18.8 | 28.6 | 0.0 | 38.2 | 14.6 | 9.2 | 10.7 | 59.8 | 66.0 | 14.3 | 16.7 | 38.8 | 41.3 | 0.0 | 53.3 |
| Qwen1.5-Chat | 14B | 35.7 | 50.0 | 54.9 | 61.9 | 19.6 | 51.3 | 37.8 | 6.6 | 27.9 | 16.7 | 30.6 | 30.4 | 46.3 | 51.1 | 28.6 | 28.6 | 59.2 | 50.0 | 54.2 | 53.3 |
| CodeLlama-Instruct | 13B | 78.6 | 57.3 | 75.6 | 79.8 | 48.2 | 56.3 | 48.0 | 52.6 | 54.4 | 41.7 | 43.9 | 46.4 | 68.3 | 68.1 | 26.2 | 21.4 | 52.0 | 61.3 | 48.6 | 66.7 |
| Yi-1.5-Chat | 9B | 35.7 | 63.4 | 65.9 | 84.5 | 39.3 | 61.3 | 66.3 | 38.2 | 48.5 | 27.1 | 68.4 | 35.7 | 59.8 | 85.1 | 38.1 | 21.4 | 77.6 | 58.8 | 69.4 | 75.6 |
| CodeLlama-Instruct | 7B | 78.6 | 62.2 | 73.2 | 54.8 | 30.4 | 66.3 | 48.0 | 31.6 | 47.1 | 35.4 | 52.0 | 42.9 | 61.0 | 61.7 | 33.3 | 26.2 | 22.4 | 18.6 | 61.1 | 67.8 |
| CodeQwen1.5-Chat | 7B | 35.7 | 68.3 | 63.4 | 76.2 | 67.9 | 47.5 | 78.6 | 35.5 | 72.1 | 45.8 | 57.1 | 60.7 | 73.2 | 69.1 | 31.0 | 42.9 | 90.8 | 68.8 | 69.4 | 88.9 |
| Magicoder-S-DS | 7B | 71.4 | 67.1 | 72.0 | 84.5 | 71.4 | 77.5 | 80.6 | 69.7 | 79.4 | 62.5 | 85.7 | 85.7 | 75.6 | 96.8 | 52.4 | 40.5 | 94.9 | 72.5 | 73.6 | 74.4 |
| **MCODER** | 7B | 85.7 | 74.4 | 82.9 | 90.5 | 69.6 | 91.3 | 78.6 | 69.7 | 77.9 | 70.8 | 67.3 | 75.0 | 80.5 | 83.0 | 45.2 | 28.6 | 95.9 | 81.3 | 54.2 | 94.4 |

| Method | Kotlin | Lua | MD | Pascal | Perl | PHP | Power | Python | R | Racket | Ruby | Rust | Scala | Scheme | Shell | Swift | Tcl | TS | VB | VimL | Avg$_{all}$ |
|---|---|---|---|---|---|---|---|---|---|---|---|---|---|---|---|---|---|---|---|---|---|
| GPT-4 Turbo (231106) | 83.3 | 80.0 | 28.6 | 70.7 | 91.4 | 86.2 | 92.2 | 86.7 | 87.0 | 88.7 | 89.5 | 81.5 | 87.8 | 85.3 | 87.8 | 86.1 | 85.9 | 77.5 | 76.9 | 63.2 | 82.7 |
| GPT-3.5 Turbo (240125) | 64.3 | 58.8 | 21.4 | 20.7 | 52.9 | 20.2 | 65.6 | 84.4 | 23.9 | 41.9 | 47.4 | 27.2 | 50.0 | 39.7 | 54.9 | 63.9 | 57.7 | 46.3 | 28.2 | 80.3 | 43.6 |
| DeepSeekCoder-Instruct | 73.8 | 71.3 | 14.3 | 68.3 | 65.7 | 81.9 | 86.7 | 83.3 | 79.3 | 59.7 | 86.8 | 75.0 | 84.1 | 72.1 | 80.5 | 79.2 | 83.3 | 57.5 | 70.5 | 67.1 | 72.4 |
| OCTOCODER | 70.2 | 50.0 | 3.6 | 15.9 | 32.9 | 0.0 | 28.9 | 78.9 | 2.2 | 38.7 | 15.8 | 0.0 | 56.1 | 41.2 | 48.8 | 27.8 | 37.2 | 67.5 | 59.0 | 69.7 | 39.2 |
| StarCoder2-instruct-V0.1 | 85.7 | 61.3 | 0.0 | 75.6 | 80.0 | 93.6 | 87.8 | 88.0 | 66.1 | 73.7 | 89.1 | 39.0 | 75.0 | 78.0 | 54.2 | 62.8 | 82.5 | 64.1 | 72.4 | | 71.4 |
| WizardCoder-V1.0 | 40.5 | 35.0 | 0.0 | 28.0 | 28.6 | 29.8 | 40.0 | 52.2 | 48.9 | 1.6 | 28.9 | 18.5 | 28.0 | 0.0 | 17.1 | 33.3 | 37.2 | 35.0 | 51.3 | 65.8 | 31.6 |
| Qwen1.5-Chat | 51.2 | 51.3 | 7.1 | 36.6 | 51.4 | 62.8 | 56.7 | 63.3 | 63.0 | 29.0 | 52.6 | 50.0 | 17.1 | 26.5 | 50.0 | 63.9 | 43.6 | 46.3 | 43.6 | 36.8 | 44.7 |
| CodeLlama-Instruct | 58.3 | 66.3 | 0.0 | 41.5 | 55.7 | 54.3 | 76.7 | 77.8 | 69.6 | 51.6 | 60.5 | 58.7 | 64.6 | 47.1 | 67.1 | 63.9 | 62.8 | 62.5 | 66.7 | 68.4 | 59.3 |
| Yi-1.5-Chat | 75.0 | 66.3 | 17.9 | 53.7 | 64.3 | 83.0 | 76.7 | 78.9 | 76.1 | 38.7 | 67.1 | 65.2 | 45.1 | 42.6 | 65.9 | 75.0 | 62.8 | 57.5 | 60.3 | 53.9 | 62.2 |
| CodeLlama-Instruct | 61.9 | 41.3 | 0.0 | 19.5 | 57.1 | 22.3 | 42.2 | 73.3 | 62.0 | 35.5 | 67.1 | 46.7 | 72.0 | 33.8 | 73.2 | 33.3 | 61.5 | 43.8 | 61.5 | 53.2 | 49.9 |
| CodeQwen1.5-Chat | 78.6 | 57.5 | 0.0 | 70.7 | 80.0 | 83.0 | 82.2 | 84.4 | 68.5 | 66.1 | 75.0 | 80.4 | 67.1 | 50.0 | 89.0 | 77.8 | 80.8 | 63.8 | 48.7 | 65.8 | 68.6 |
| Magicoder-S-DS | 82.1 | 77.5 | 7.1 | 75.6 | 91.4 | 90.4 | 90.0 | 87.8 | 84.8 | 66.1 | 85.5 | 78.3 | 89.0 | 79.4 | 91.5 | 86.1 | 83.3 | 70.0 | 76.9 | 39.5 | 78.4 |
| **MCODER** | 82.1 | 80.0 | 0.0 | 79.3 | 97.1 | 86.2 | 88.9 | 82.2 | 88.0 | 77.4 | 77.6 | 82.6 | 80.5 | 72.1 | 91.5 | 80.6 | 84.6 | 82.5 | 73.1 | 67.1 | 78.9 |

**Multi-line Completion**

| Method | Size | AWK | C | C++ | C# | Clisp | Coffee | Dart | Elisp | Elixir | Erlang | Fortran | F# | Go | Groovy | Haskell | Html | Java | JS | Json | Julia |
|---|---|---|---|---|---|---|---|---|---|---|---|---|---|---|---|---|---|---|---|---|---|
| GPT-4 Turbo (231106) | 🔒 | 78.6 | 69.5 | 69.5 | 83.3 | 71.4 | 96.3 | 68.4 | 77.6 | 75.0 | 54.2 | 67.3 | 64.3 | 64.6 | 92.6 | 64.3 | 33.3 | 84.7 | 71.3 | 90.3 | 83.3 |
| GPT-3.5 Turbo (240125) | 🔒 | 42.9 | 58.5 | 54.9 | 70.2 | 28.6 | 65.0 | 53.1 | 39.5 | 22.1 | 4.2 | 20.4 | 35.7 | 62.2 | 92.6 | 26.2 | 31.0 | 75.5 | 58.8 | 43.1 | 60.0 |
| DeepSeekCoder-Instruct | 33B | 71.4 | 61.0 | 62.2 | 75.0 | 37.5 | 51.3 | 50.0 | 42.1 | 50.0 | 27.1 | 69.4 | 42.9 | 47.6 | 87.2 | 26.2 | 31.0 | 80.6 | 70.0 | 83.3 | 65.6 |
| OCTOCODER | 16B | 35.7 | 35.4 | 45.1 | 61.9 | 23.2 | 30.0 | 30.6 | 7.9 | 5.9 | 0.0 | 0.0 | 25.0 | 37.8 | 60.6 | 0.0 | 0.0 | 71.4 | 47.5 | 6.9 | 1.1 |
| StarCoder2-instruct-V0.1 | 15B | 7.1 | 59.8 | 67.1 | 78.6 | 53.6 | 3.8 | 58.2 | 52.6 | 57.4 | 39.6 | 62.2 | 39.3 | 56.1 | 89.4 | 28.6 | 0.0 | 81.6 | 71.3 | 9.7 | 65.6 |
| WizardCoder-V1.0 | 15B | 42.9 | 28.0 | 31.7 | 34.5 | 3.6 | 13.8 | 21.4 | 0.0 | 20.6 | 10.4 | 11.2 | 10.7 | 39.0 | 61.7 | 9.5 | 14.3 | 24.5 | 20.0 | 1.4 | 28.9 |
| Qwen1.5-Chat | 14B | 7.1 | 40.2 | 41.5 | 42.9 | 7.1 | 26.3 | 27.6 | 5.3 | 14.7 | 4.2 | 23.5 | 12.5 | 36.6 | 47.9 | 11.9 | 14.3 | 49.0 | 42.5 | 41.7 | 45.6 |
| CodeLlama-Instruct | 13B | 50.0 | 39.0 | 50.0 | 63.1 | 25.0 | 40.0 | 31.6 | 13.2 | 22.1 | 18.8 | 16.3 | 23.2 | 35.4 | 50.0 | 19.0 | 7.1 | 41.8 | 45.0 | 54.2 | 45.6 |
| Yi-1.5-Chat | 9B | 35.7 | 47.6 | 48.8 | 64.3 | 14.3 | 52.5 | 42.9 | 18.4 | 20.6 | 16.7 | 51.0 | 25.0 | 39.0 | 73.4 | 23.8 | 16.7 | 61.2 | 55.0 | 75.0 | 60.0 |
| CodeLlama-Instruct | 7B | 28.6 | 36.6 | 45.1 | 42.9 | 17.9 | 46.3 | 16.3 | 17.1 | 22.1 | 25.0 | 22.4 | 28.6 | 37.8 | 43.6 | 23.8 | 31.0 | 24.5 | 12.5 | 73.6 | 34.4 |
| CodeQwen1.5-Chat | 7B | 0.0 | 52.4 | 53.7 | 73.8 | 30.4 | 25.0 | 58.2 | 34.2 | 42.6 | 43.8 | 51.0 | 42.9 | 58.5 | 67.0 | 31.0 | 35.7 | 78.6 | 71.3 | 72.2 | 68.9 |
| Magicoder-S-DS | 7B | 21.4 | 64.6 | 64.6 | 81.0 | 51.8 | 55.0 | 59.2 | 52.6 | 60.3 | 45.8 | 69.4 | 66.1 | 62.2 | 91.5 | 35.7 | 16.7 | 78.6 | 65.0 | 69.4 | 62.2 |
| **MCODER** | 7B | 21.4 | 57.3 | 53.7 | 78.6 | 42.9 | 71.3 | 56.1 | 50.0 | 57.4 | 47.9 | 45.9 | 51.8 | 56.1 | 80.9 | 23.8 | 23.8 | 80.6 | 75.0 | 62.5 | 72.2 |

| Method | Kotlin | Lua | MD | Pascal | Perl | PHP | Power | Python | R | Racket | Ruby | Rust | Scala | Scheme | Shell | Swift | Tcl | TS | VB | VimL | Avg$_{all}$ |
|---|---|---|---|---|---|---|---|---|---|---|---|---|---|---|---|---|---|---|---|---|---|
| GPT-4 Turbo (231106) | 84.5 | 66.3 | 32.1 | 67.1 | 84.3 | 85.1 | 94.4 | 83.3 | 80.4 | 64.5 | 78.9 | 81.5 | 84.1 | 86.8 | 90.2 | 81.9 | 78.2 | 75.0 | 75.6 | 52.6 | 76.6 |
| GPT-3.5 Turbo (240125) | 70.2 | 63.8 | 17.9 | 51.2 | 67.1 | 67.1 | 11.7 | 63.3 | 88.9 | 32.6 | 30.6 | 42.1 | 34.8 | 37.8 | 44.1 | 69.5 | 72.2 | 65.4 | 55.1 | 53.9 | 51.6 |
| DeepSeekCoder-Instruct | 69.0 | 58.8 | 21.4 | 50.0 | 60.0 | 73.4 | 82.2 | 72.2 | 65.2 | 40.3 | 64.5 | 58.7 | 74.4 | 47.1 | 81.7 | 68.1 | 66.7 | 57.5 | 66.7 | 34.2 | 61.8 |
| OCTOCODER | 53.6 | 30.0 | 3.6 | 3.7 | 35.7 | 0.0 | 26.7 | 52.2 | 4.3 | 27.4 | 2.6 | 0.0 | 34.1 | 23.5 | 32.9 | 33.3 | 25.6 | 45.0 | 33.3 | 35.5 | 27.1 |
| StarCoder2-instruct-V0.1 | 76.2 | 48.8 | 0.0 | 64.3 | 80.9 | 76.7 | 75.6 | 77.2 | 48.4 | 51.3 | 64.1 | 50.0 | 61.8 | 68.3 | 47.2 | 50.0 | 65.0 | 60.3 | 47.4 | | 58.3 |
| WizardCoder-V1.0 | 46.4 | 15.0 | 0.0 | 7.3 | 30.0 | 31.9 | 22.2 | 38.9 | 33.7 | 0.0 | 23.7 | 16.3 | 14.6 | 0.0 | 15.9 | 19.4 | 23.1 | 23.8 | 37.2 | 14.5 | 22.8 |
| Qwen1.5-Chat | 36.9 | 30.0 | 3.6 | 19.5 | 24.3 | 48.9 | 34.4 | 42.2 | 39.1 | 8.1 | 31.6 | 29.3 | 12.2 | 10.3 | 32.9 | 47.2 | 21.8 | 38.8 | 24.4 | 19.7 | 29.9 |
| CodeLlama-Instruct | 47.6 | 37.5 | 0.0 | 28.0 | 31.4 | 43.6 | 60.0 | 46.7 | 42.4 | 22.6 | 39.5 | 41.3 | 35.4 | 19.1 | 40.2 | 48.6 | 41.0 | 42.5 | 47.4 | 34.2 | 39.0 |
| Yi-1.5-Chat | 52.4 | 50.0 | 3.6 | 37.8 | 52.9 | 67.0 | 60.0 | 63.3 | 53.3 | 29.0 | 42.1 | 39.1 | 39.0 | 17.6 | 57.3 | 69.4 | 38.5 | 56.3 | 50.0 | 26.3 | 46.6 |
| CodeLlama-Instruct | 44.0 | 20.0 | 0.0 | 14.6 | 30.0 | 8.5 | 38.9 | 45.6 | 27.2 | 22.6 | 31.6 | 27.2 | 40.2 | 14.7 | 46.3 | 38.9 | 41.0 | 27.5 | 44.9 | 28.9 | 31.4 |
| CodeQwen1.5-Chat | 69.0 | 51.3 | 3.6 | 47.6 | 61.4 | 70.2 | 76.7 | 57.8 | 62.0 | 46.8 | 44.7 | 67.4 | 69.5 | 38.2 | 76.8 | 70.8 | 60.3 | 56.3 | 46.2 | 40.8 | 56.3 |
| Magicoder-S-DS | 77.4 | 57.5 | 10.7 | 64.6 | 68.6 | 72.3 | 84.4 | 76.7 | 70.7 | 51.6 | 64.5 | 64.1 | 82.9 | 58.8 | 75.6 | 73.6 | 66.7 | 62.5 | 64.1 | 23.7 | 65.4 |
| **MCODER** | 66.7 | 56.3 | 0.0 | 58.5 | 57.1 | 71.3 | 80.0 | 66.7 | 64.1 | 50.0 | 57.9 | 71.7 | 63.4 | 54.4 | 80.5 | 63.9 | 57.7 | 60.0 | 66.7 | 32.9 | 60.7 |

**Span Completion**

| Method | Size | AWK | C | C++ | C# | Clisp | Coffee | Dart | Elisp | Elixir | Erlang | Fortran | F# | Go | Groovy | Haskell | Html | Java | JS | Json | Julia |
|---|---|---|---|---|---|---|---|---|---|---|---|---|---|---|---|---|---|---|---|---|---|
| GPT-4 Turbo (231106) | 🔒 | 80.0 | 65.0 | 67.0 | 84.0 | 74.0 | 88.0 | 63.7 | 71.0 | 74.0 | 68.0 | 69.0 | 78.0 | 67.0 | 94.0 | 80.0 | 49.0 | 91.5 | 65.0 | 83.0 | 87.0 |
| GPT-3.5 Turbo (240125) | 🔒 | 39.0 | 41.0 | 51.0 | 70.0 | 55.0 | 67.0 | 58.8 | 43.0 | 31.0 | 9.0 | 27.0 | 55.0 | 57.0 | 85.0 | 14.0 | 30.0 | 58.5 | 47.0 | 30.0 | 48.0 |
| DeepSeekCoder-Instruct | 33B | 52.0 | 55.0 | 59.0 | 70.0 | 58.0 | 57.0 | 48.0 | 45.0 | 57.0 | 58.0 | 65.0 | 61.0 | 48.0 | 86.0 | 53.0 | 42.0 | 70.8 | 54.0 | 61.0 | 68.0 |
| OCTOCODER | 16B | 29.0 | 41.0 | 46.0 | 51.0 | 41.0 | 29.0 | 31.4 | 19.0 | 9.0 | 0.0 | 0.0 | 44.0 | 40.0 | 52.0 | 0.0 | 4.0 | 57.5 | 39.0 | 12.0 | 1.0 |
| StarCoder2-instruct-V0.1 | 15B | 34.0 | 63.0 | 67.0 | 75.0 | 62.0 | 2.0 | 49.0 | 48.0 | 56.0 | 58.0 | 62.0 | 59.0 | 59.0 | 80.0 | 59.0 | 0.0 | 82.1 | 59.0 | 10.0 | 76.0 |
| WizardCoder-V1.0 | 15B | 19.0 | 24.0 | 40.0 | 35.0 | 6.0 | 11.0 | 6.9 | 3.0 | 6.0 | 11.0 | 4.0 | 7.0 | 44.0 | 57.0 | 26.0 | 9.0 | 16.0 | 13.0 | 0.0 | 14.0 |
| Qwen1.5-Chat | 14B | 27.0 | 43.0 | 47.0 | 43.0 | 21.0 | 10.0 | 26.5 | 12.0 | 25.0 | 31.0 | 22.0 | 25.0 | 37.0 | 38.0 | 39.0 | 17.0 | 26.4 | 43.0 | 23.0 | 43.0 |
| CodeLlama-Instruct | 13B | 47.0 | 36.0 | 51.0 | 56.0 | 40.0 | 29.0 | 25.5 | 22.0 | 36.0 | 45.0 | 35.0 | 40.0 | 39.0 | 46.0 | 36.0 | 28.0 | 39.6 | 44.0 | 43.0 | 44.0 |
| Yi-1.5-Chat | 9B | 37.0 | 56.0 | 60.0 | 66.0 | 27.0 | 47.0 | 51.0 | 23.0 | 33.0 | 40.0 | 48.0 | 30.0 | 43.0 | 72.0 | 51.0 | 25.0 | 64.6 | 56.3 | 57.0 | 61.0 |
| CodeLlama-Instruct | 7B | 25.0 | 43.0 | 51.0 | 53.0 | 43.0 | 11.0 | 39.0 | 27.5 | 20.0 | 38.0 | 41.0 | 35.0 | 41.0 | 43.0 | 25.0 | 26.4 | 29.0 | 58.0 | 55.0 | 43.0 |
| CodeQwen1.5-Chat | 7B | 41.0 | 57.0 | 59.0 | 65.0 | 54.0 | 22.0 | 46.1 | 34.0 | 54.0 | 56.0 | 58.0 | 62.0 | 57.0 | 74.0 | 52.0 | 39.0 | 79.2 | 55.0 | 59.0 | 76.0 |
| Magicoder-S-DS | 7B | 48.0 | 59.0 | 67.0 | 74.0 | 63.0 | 63.0 | 57.8 | 50.0 | 64.0 | 72.0 | 79.0 | 79.0 | 77.0 | 94.0 | 60.0 | 40.0 | 81.1 | 59.0 | 55.0 | 64.0 |
| **MCODER** | 7B | 43.0 | 64.0 | 66.0 | 77.0 | 62.0 | 76.0 | 52.9 | 44.0 | 61.0 | 71.0 | 70.0 | 69.0 | 55.0 | 84.0 | 65.0 | 25.0 | 83.0 | 59.0 | 54.0 | 83.0 |

| Method | Kotlin | Lua | MD | Pascal | Perl | PHP | Power | Python | R | Racket | Ruby | Rust | Scala | Scheme | Shell | Swift | Tcl | TS | VB | VimL | Avg$_{all}$ |
|---|---|---|---|---|---|---|---|---|---|---|---|---|---|---|---|---|---|---|---|---|---|
| GPT-4 Turbo (231106) | 90.0 | 66.0 | 34.0 | 59.0 | 92.0 | 80.0 | 87.0 | 80.0 | 79.0 | 80.0 | 81.1 | 82.0 | 76.0 | 82.0 | 86.0 | 71.0 | 67.0 | 73.0 | 49.0 | | 75.0 |
| GPT-3.5 Turbo (240125) | 68.0 | 61.0 | 25.0 | 45.0 | 68.0 | 62.0 | 80.0 | 13.0 | 56.0 | 58.0 | 36.0 | 24.0 | 26.4 | 39.0 | 47.0 | 72.0 | 70.0 | 57.0 | 60.0 | 58.0 | 48.1 |
| DeepSeekCoder-Instruct | 69.0 | 58.0 | 16.0 | 50.0 | 54.0 | 65.0 | 72.0 | 66.0 | 61.0 | 58.0 | 63.0 | 56.6 | 73.0 | 53.0 | 64.0 | 68.0 | 59.0 | 60.0 | 72.0 | 43.0 | 59.5 |
| OCTOCODER | 47.0 | 47.0 | 3.0 | 10.0 | 24.0 | 0.0 | 31.0 | 51.0 | 0.0 | 37.0 | 2.0 | 0.0 | 37.0 | 25.0 | 22.0 | 38.0 | 26.0 | 44.0 | 42.0 | 30.0 | 26.6 |
| StarCoder2-instruct-V0.1 | 68.0 | 41.0 | 0.0 | 58.0 | 74.0 | 69.0 | 67.0 | 71.0 | 67.0 | 67.0 | 58.0 | 68.9 | 43.0 | 60.0 | 60.0 | 56.0 | 51.0 | 63.0 | 59.0 | 51.0 | 55.4 |
| WizardCoder-V1.0 | 25.0 | 11.0 | 0.0 | 10.0 | 38.0 | 12.0 | 19.0 | 26.0 | 27.0 | 1.0 | 8.0 | 10.4 | 12.0 | 0.0 | 7.0 | 6.0 | 20.0 | 15.0 | 47.0 | 22.0 | 17.0 |
| Qwen1.5-Chat | 33.0 | 47.0 | 8.0 | 30.0 | 37.0 | 44.0 | 38.0 | 45.0 | 46.0 | 25.0 | 41.0 | 32.1 | 15.0 | 22.0 | 35.0 | 48.0 | 28.0 | 46.0 | 38.0 | 23.0 | 32.0 |
| CodeLlama-Instruct | 45.0 | 44.0 | 0.0 | 27.0 | 51.0 | 38.0 | 49.0 | 47.0 | 44.0 | 38.0 | 48.0 | 29.2 | 47.0 | 33.0 | 33.0 | 55.0 | 37.0 | 45.0 | 49.0 | 44.0 | 40.5 |
| Yi-1.5-Chat | 56.0 | 52.0 | 12.0 | 49.0 | 57.0 | 61.0 | 63.0 | 64.0 | 56.0 | 33.0 | 57.0 | 37.7 | 28.0 | 30.0 | 43.0 | 66.0 | 38.0 | 53.0 | 37.0 | | 47.3 |
| CodeLlama-Instruct | 39.0 | 30.0 | 1.0 | 24.0 | 41.0 | 15.0 | 41.0 | 46.0 | 40.0 | 38.0 | 41.0 | 28.3 | 48.0 | 22.0 | 48.0 | 53.0 | 39.0 | 37.0 | 53.0 | 35.0 | 37.5 |
| CodeQwen1.5-Chat | 70.0 | 52.0 | 15.0 | 49.0 | 76.0 | 62.0 | 59.0 | 66.0 | 62.0 | 58.0 | 64.0 | 70.8 | 57.0 | 48.0 | 70.0 | 72.0 | 59.0 | 58.0 | 44.0 | 45.0 | 56.4 |
| Magicoder-S-DS | 75.0 | 59.0 | 16.0 | 53.0 | 80.0 | 67.0 | 72.0 | 69.0 | 69.0 | 64.0 | 72.0 | 58.5 | 78.0 | 59.0 | 77.0 | 78.0 | 69.0 | 61.0 | 70.0 | 30.0 | 64.8 |
| **MCODER** | 67.0 | 63.0 | 0.0 | 58.0 | 73.0 | 66.0 | 69.0 | 67.0 | 64.0 | 63.0 | 68.0 | 72.6 | 56.0 | 61.0 | 71.0 | 79.0 | 63.0 | 64.0 | 70.0 | 43.0 | 62.6 |

**Span Completion Light**

| Method | Size | AWK | C | C++ | C# | Clisp | Coffee | Dart | Elisp | Elixir | Erlang | Fortran | F# | Go | Groovy | Haskell | Html | Java | JS | Json | Julia |
|---|---|---|---|---|---|---|---|---|---|---|---|---|---|---|---|---|---|---|---|---|---|
| GPT-4 Turbo (231106) | 🔒 | 76.0 | 66.0 | 70.0 | 84.0 | 80.0 | 94.0 | 66.7 | 74.0 | 76.0 | 60.0 | 72.0 | 72.0 | 66.0 | 90.0 | 86.0 | 46.0 | 88.7 | 62.0 | 88.0 | 86.0 |
| GPT-3.5 Turbo (240125) | 🔒 | 40.0 | 52.0 | 46.0 | 60.0 | 62.0 | 68.0 | 49.0 | 40.0 | 34.0 | 20.0 | 38.0 | 62.0 | 58.0 | 92.0 | 28.0 | 44.0 | 52.8 | 60.0 | 28.0 | 32.0 |
| DeepSeekCoder-Instruct | 33B | 60.0 | 54.0 | 60.0 | 76.0 | 50.0 | 56.0 | 35.3 | 44.0 | 54.0 | 60.0 | 58.0 | 66.0 | 60.0 | 88.0 | 44.0 | 38.0 | 73.6 | 64.0 | 66.0 | 64.0 |
| OCTOCODER | 16B | 22.0 | 40.0 | 40.0 | 58.0 | 42.0 | 22.0 | 27.5 | 20.0 | 8.0 | 0.0 | 0.0 | 46.0 | 52.0 | 0.0 | 0.0 | 2.0 | 54.7 | 44.0 | 12.0 | 0.0 |
| StarCoder2-instruct-V0.1 | 15B | 34.0 | 48.0 | 60.0 | 74.0 | 62.0 | 4.0 | 47.1 | 44.0 | 64.0 | 56.0 | 58.0 | 68.0 | 56.0 | 82.0 | 60.0 | 0.0 | 73.6 | 66.0 | 6.0 | 66.0 |
| WizardCoder-V1.0 | 15B | 24.0 | 20.0 | 30.0 | 36.0 | 10.0 | 14.0 | 9.8 | 2.0 | 4.0 | 10.0 | 6.0 | 2.0 | 44.0 | 48.0 | 24.0 | 8.0 | 28.3 | 14.0 | 0.0 | 22.0 |
| Qwen1.5-Chat | 14B | 40.0 | 38.0 | 44.0 | 52.0 | 16.0 | 20.0 | 21.6 | 14.0 | 28.0 | 24.0 | 18.0 | 24.0 | 36.0 | 38.0 | 38.0 | 14.0 | 37.7 | 44.0 | 44.0 | 56.0 |
| CodeLlama-Instruct | 13B | 48.0 | 46.0 | 42.0 | 58.0 | 34.0 | 20.0 | 23.5 | 30.0 | 36.0 | 40.0 | 30.0 | 38.0 | 42.0 | 36.0 | 36.0 | 26.0 | 49.1 | 44.0 | 44.0 | 56.0 |
| Yi-1.5-Chat | 9B | 36.0 | 52.0 | 58.0 | 68.0 | 38.0 | 54.0 | 39.2 | 24.0 | 26.0 | 34.0 | 52.0 | 24.0 | 50.0 | 70.0 | 46.0 | 34.0 | 71.7 | 52.0 | 58.0 | 56.0 |
| CodeLlama-Instruct | 7B | 30.0 | 38.0 | 46.0 | 44.0 | 40.0 | 32.0 | 19.6 | 28.0 | 34.0 | 38.0 | 32.0 | 42.0 | 50.0 | 34.0 | 36.0 | 20.0 | 32.1 | 38.0 | 50.0 | 38.0 |
| CodeQwen1.5-Chat | 7B | 48.0 | 52.0 | 60.0 | 78.0 | 60.0 | 22.0 | 47.1 | 32.0 | 54.0 | 52.0 | 48.0 | 30.0 | 48.0 | 76.0 | 52.0 | 30.0 | 88.7 | 68.0 | 68.0 | 68.0 |
| Magicoder-S-DS | 7B | 54.0 | 54.0 | 66.0 | 80.0 | 60.0 | 60.0 | 56.9 | 52.0 | 66.0 | 70.0 | 72.0 | 78.0 | 66.0 | 94.0 | 60.0 | 30.0 | 83.0 | 58.0 | 56.0 | 62.0 |
| **MCODER** | 7B | 58.0 | 56.0 | 68.0 | 82.0 | 64.0 | 76.0 | 58.8 | 40.0 | 62.0 | 76.0 | 62.0 | 74.0 | 56.0 | 88.0 | 64.0 | 18.0 | 88.7 | 70.0 | 56.0 | 72.0 |

| Method | Kotlin | Lua | MD | Pascal | Perl | PHP | Power | Python | R | Racket | Ruby | Rust | Scala | Scheme | Shell | Swift | Tcl | TS | VB | VimL | Avg$_{all}$ |
|---|---|---|---|---|---|---|---|---|---|---|---|---|---|---|---|---|---|---|---|---|---|
| GPT-4 Turbo (231106) | 84.0 | 74.0 | 44.0 | 70.0 | 90.0 | 74.0 | 88.0 | 90.0 | 86.0 | 82.0 | 88.0 | 81.1 | 80.0 | 76.0 | 80.0 | 82.0 | 82.0 | 64.0 | 80.0 | 58.0 | 76.5 |
| GPT-3.5 Turbo (240125) | 60.0 | 56.0 | 24.0 | 52.0 | 70.0 | 24.0 | 56.0 | 82.0 | 36.0 | 60.0 | 36.0 | 43.4 | 48.0 | 46.0 | 70.0 | 62.0 | 58.0 | 50.0 | 64.0 | 52.0 | 50.4 |
| DeepSeekCoder-Instruct | 58.0 | 60.0 | 32.0 | 52.0 | 68.0 | 60.0 | 80.0 | 66.0 | 72.0 | 56.0 | 66.0 | 54.7 | 76.0 | 54.0 | 66.0 | 70.0 | 68.0 | 56.0 | 68.0 | 40.0 | 60.9 |
| OCTOCODER | 42.0 | 46.0 | 0.0 | 8.0 | 40.0 | 0.0 | 32.0 | 52.0 | 2.0 | 32.0 | 4.0 | 0.0 | 34.0 | 34.0 | 28.0 | 38.0 | 28.0 | 50.0 | 54.0 | 36.0 | 27.0 |
| StarCoder2-instruct-V0.1 | 70.0 | 54.0 | 0.0 | 60.0 | 78.0 | 78.0 | 78.0 | 80.0 | 70.0 | 58.0 | 70.0 | 71.7 | 36.0 | 54.0 | 52.0 | 60.0 | 54.0 | 66.0 | 62.0 | 58.0 | 55.5 |
| WizardCoder-V1.0 | 38.0 | 8.0 | 0.0 | 8.0 | 42.0 | 14.0 | 20.0 | 26.0 | 28.0 | 2.0 | 6.0 | 3.8 | 18.0 | 0.0 | 6.0 | 12.0 | 10.0 | 12.0 | 24.0 | 24.0 | 17.6 |
| Qwen1.5-Chat | 26.0 | 46.0 | 6.0 | 26.0 | 32.0 | 46.0 | 40.0 | 44.0 | 44.0 | 32.0 | 46.0 | 32.1 | 16.0 | 16.0 | 34.0 | 50.0 | 20.0 | 40.0 | 44.0 | 24.0 | 32.4 |
| CodeLlama-Instruct | 44.0 | 52.0 | 0.0 | 26.0 | 44.0 | 38.0 | 58.0 | 48.0 | 34.0 | 38.0 | 46.0 | 37.7 | 40.0 | 36.0 | 58.0 | 40.0 | 38.0 | 58.0 | 42.0 | | 39.9 |
| Yi-1.5-Chat | 52.0 | 56.0 | 8.0 | 50.0 | 56.0 | 70.0 | 62.0 | 62.0 | 62.0 | 40.0 | 58.0 | 50.9 | 32.0 | 26.0 | 56.0 | 60.0 | 38.0 | 56.0 | 68.0 | 36.0 | 48.6 |
| CodeLlama-Instruct | 34.0 | 36.0 | 2.0 | 22.0 | 44.0 | 12.0 | 42.0 | 48.0 | 28.0 | 38.0 | 40.0 | 34.0 | 44.0 | 32.0 | 42.0 | 54.0 | 36.0 | 40.0 | 58.0 | 36.0 | 36.3 |
| CodeQwen1.5-Chat | 70.0 | 60.0 | 22.0 | 54.0 | 76.0 | 68.0 | 72.0 | 66.0 | 64.0 | 64.0 | 66.0 | 62.0 | 46.0 | 76.0 | 72.0 | 56.0 | 22.0 | 40.0 | 34.0 | | 55.7 |
| Magicoder-S-DS | 74.0 | 66.0 | 18.0 | 66.0 | 84.0 | 72.0 | 78.0 | 76.0 | 72.0 | 58.0 | 64.0 | 54.0 | 78.0 | 66.0 | 74.0 | 78.0 | 72.0 | 56.0 | 68.0 | 34.0 | 65.4 |
| **MCODER** | 62.0 | 68.0 | 4.0 | 62.0 | 72.0 | 74.0 | 70.0 | 62.0 | 66.0 | 66.0 | 70.0 | 66.0 | 68.0 | 48.0 | 74.0 | 74.0 | 56.0 | 62.0 | 70.0 | 40.0 | 63.1 |

state-of-the-art models and open-source models across nearly all programming languages. Notably, GPT-4o and GPT-4 Turbo lead the benchmark with substantial performance margins over other models. The MCEVAL apart from previous benchmarks (such as HumanEval), where various open models have achieved comparable or superior performance. The results indicate that MCODER

Table 5: Pass@1 (%) scores of different models for multilingual code explanation tasks on MCEVAL. "Avg$_{all}$" represents the average scores of all code languages.

| Method | Size | AWK | C | C++ | C# | Clisp | Coffee | Dart | Elisp | Elixir | Erlang | Fortran | F# | Go | Groovy | Haskell | Html | Java | JS | Json | Julia |
|---|---|---|---|---|---|---|---|---|---|---|---|---|---|---|---|---|---|---|---|---|---|
| GPT-4o (240513) | 🔒 | 74.0 | 68.0 | 72.0 | 72.0 | 78.0 | 76.0 | 54.9 | 60.0 | 66.0 | 38.0 | 62.0 | 78.0 | 74.0 | 88.0 | 86.0 | 16.0 | 79.2 | 68.0 | 24.0 | 76.0 |
| GPT-4 Turbo (231106) | 🔒 | 84.0 | 70.0 | 72.0 | 52.0 | 60.0 | 76.0 | 54.9 | 50.0 | 64.0 | 42.0 | 50.0 | 78.0 | 68.0 | 88.0 | 76.0 | 10.0 | 81.1 | 74.0 | 18.0 | 70.0 |
| GPT-3.5 Turbo (240125) | 🔒 | 8.0 | 68.0 | 64.0 | 62.0 | 56.0 | 66.0 | 29.4 | 26.0 | 42.0 | 34.0 | 28.0 | 56.0 | 54.0 | 84.0 | 54.0 | 6.0 | 73.6 | 58.0 | 6.0 | 62.0 |
| Yi-Large-Turbo | 🔒 | 76.0 | 52.0 | 48.0 | 50.0 | 60.0 | 60.0 | 25.5 | 44.0 | 42.0 | 44.0 | 42.0 | 60.0 | 54.0 | 74.0 | 68.0 | 14.0 | 47.2 | 48.0 | 26.0 | 60.0 |
| DeepSeekCoder-Instruct | 33B | 70.0 | 62.0 | 66.0 | 78.0 | 56.0 | 68.0 | 45.1 | 44.0 | 58.0 | 40.0 | 44.0 | 64.0 | 56.0 | 88.0 | 52.0 | 8.0 | 67.9 | 52.0 | 32.0 | 70.0 |
| OCTOCODER | 16B | 32.0 | 32.0 | 32.0 | 26.0 | 48.0 | 4.0 | 5.9 | 22.0 | 32.0 | 34.0 | 22.0 | 16.0 | 52.0 | 42.0 | 34.0 | 2.0 | 37.7 | 36.0 | 8.0 | 28.0 |
| Qwen1.5-Chat | 14B | 50.0 | 52.0 | 30.0 | 42.0 | 36.0 | 26.0 | 23.5 | 28.0 | 24.0 | 12.0 | 14.0 | 34.0 | 44.0 | 28.0 | 46.0 | 8.0 | 35.8 | 44.0 | 14.0 | 42.0 |
| CodeLlama-Instruct | 13B | 32.0 | 42.0 | 34.0 | 44.0 | 44.0 | 0.0 | 7.8 | 16.0 | 30.0 | 38.0 | 16.0 | 30.0 | 32.0 | 32.0 | 40.0 | 10.0 | 28.3 | 36.0 | 10.0 | 24.0 |
| Yi-1.5-Chat | 9B | 38.0 | 62.0 | 60.0 | 58.0 | 38.0 | 38.0 | 29.4 | 14.0 | 50.0 | 26.0 | 36.0 | 20.0 | 52.0 | 68.0 | 44.0 | 10.0 | 67.9 | 46.0 | 22.0 | 66.0 |
| CodeLlama-Instruct | 7B | 34.0 | 34.0 | 32.0 | 42.0 | 34.0 | 8.0 | 11.8 | 22.0 | 28.0 | 24.0 | 14.0 | 26.0 | 40.0 | 22.0 | 36.0 | 0.0 | 34.0 | 22.0 | 8.0 | 14.0 |
| CodeQwen1.5-Chat | 7B | 58.0 | 62.0 | 58.0 | 50.0 | 54.0 | 56.0 | 17.6 | 26.0 | 50.0 | 52.0 | 34.0 | 50.0 | 48.0 | 76.0 | 46.0 | 2.0 | 67.9 | 58.0 | 18.0 | 56.0 |
| Magicoder-S-DS | 7B | 58.0 | 62.0 | 56.0 | 70.0 | 58.0 | 4.0 | 37.3 | 28.0 | 50.0 | 58.0 | 54.0 | 66.0 | 58.0 | 86.0 | 60.0 | 4.0 | 84.9 | 60.0 | 2.0 | 54.0 |
| MCODER (Our Method) | 7B | 52.0 | 62.0 | 56.0 | 68.0 | 70.0 | 48.0 | 33.3 | 44.0 | 64.0 | 40.0 | 40.0 | 56.0 | 70.0 | 66.0 | 58.0 | 6.0 | 71.7 | 60.0 | 28.0 | 60.0 |

| Method | Kotlin | Lua | MD | Pascal | Perl | PHP | Power | Python | R | Racket | Ruby | Rust | Scala | Scheme | Shell | Swift | Tcl | TS | VB | VimL | Avg$_{all}$ |
|---|---|---|---|---|---|---|---|---|---|---|---|---|---|---|---|---|---|---|---|---|---|
| GPT-4o (240513) | 70.0 | 74.0 | 12.0 | 56.0 | 78.0 | 64.0 | 84.0 | 62.0 | 60.0 | 70.0 | 78.0 | 84.9 | 70.0 | 72.0 | 52.0 | 84.0 | 62.0 | 70.0 | 76.0 | 42.0 | 65.8 |
| GPT-4 Turbo (231106) | 68.0 | 64.0 | 8.0 | 56.0 | 68.0 | 68.0 | 82.0 | 66.0 | 60.0 | 62.0 | 70.0 | 66.0 | 70.0 | 58.0 | 68.0 | 90.0 | 50.0 | 72.0 | 68.0 | 50.0 | 62.6 |
| GPT-3.5 Turbo (240125) | 62.0 | 54.0 | 4.0 | 46.0 | 40.0 | 40.0 | 56.0 | 54.0 | 44.0 | 50.0 | 62.0 | 47.2 | 48.0 | 38.0 | 62.0 | 40.0 | 62.0 | 62.0 | 36.0 | 47.9 |
| Yi-Large-Turbo | 48.0 | 56.0 | 6.0 | 48.0 | 58.0 | 48.0 | 66.0 | 50.0 | 42.0 | 54.0 | 72.0 | 52.8 | 54.0 | 56.0 | 42.0 | 66.0 | 50.0 | 60.0 | 56.0 | 44.0 | 50.6 |
| DeepSeekCoder-Instruct | 56.0 | 64.0 | 6.0 | 42.0 | 52.0 | 52.0 | 68.0 | 58.0 | 52.0 | 50.0 | 70.0 | 49.1 | 64.0 | 54.0 | 34.0 | 68.0 | 56.0 | 62.0 | 66.0 | 40.0 | 55.8 |
| OCTOCODER | 24.0 | 36.0 | 0.0 | 24.0 | 14.0 | 32.0 | 44.0 | 62.0 | 26.0 | 34.0 | 10.0 | 26.4 | 30.0 | 34.0 | 40.0 | 30.0 | 24.0 | 34.0 | 38.0 | 8.0 | 29.4 |
| Qwen1.5-Chat | 30.0 | 36.0 | 6.0 | 22.0 | 38.0 | 34.0 | 34.0 | 32.0 | 24.0 | 48.0 | 66.0 | 22.6 | 18.0 | 24.0 | 16.0 | 60.0 | 20.0 | 58.0 | 44.0 | 30.0 | 32.4 |
| CodeLlama-Instruct | 26.0 | 36.0 | 0.0 | 4.0 | 28.0 | 30.0 | 36.0 | 26.0 | 20.0 | 32.0 | 28.0 | 20.8 | 32.0 | 26.0 | 38.0 | 32.0 | 42.0 | 50.0 | 24.0 |  | 29.3 |
| Yi-1.5-Chat | 50.0 | 56.0 | 4.0 | 30.0 | 42.0 | 52.0 | 62.0 | 70.0 | 52.0 | 34.0 | 74.0 | 30.2 | 50.0 | 42.0 | 24.0 | 52.0 | 30.0 | 68.0 | 36.0 | 28.0 | 43.3 |
| CodeLlama-Instruct | 24.0 | 30.0 | 0.0 | 4.0 | 32.0 | 2.0 | 30.0 | 22.0 | 14.0 | 28.0 | 28.0 | 17.0 | 32.0 | 24.0 | 20.0 | 42.0 | 28.0 | 30.0 | 40.0 | 12.0 | 23.8 |
| CodeQwen1.5-Chat | 48.0 | 46.0 | 4.0 | 50.0 | 44.0 | 40.0 | 44.0 | 50.0 | 38.0 | 40.0 | 58.0 | 47.2 | 52.0 | 44.0 | 40.0 | 70.0 | 46.0 | 62.0 | 64.0 | 28.0 | 46.4 |
| Magicoder-S-DS | 64.0 | 50.0 | 2.0 | 52.0 | 50.0 | 46.0 | 58.0 | 54.0 | 48.0 | 46.0 | 62.0 | 47.2 | 64.0 | 62.0 | 48.0 | 60.0 | 42.0 | 56.0 | 62.0 | 24.0 | 51.4 |
| MCODER (Our Method) | 62.0 | 52.0 | 0.0 | 48.0 | 46.0 | 44.0 | 62.0 | 58.0 | 38.0 | 50.0 | 74.0 | 67.9 | 32.0 | 46.0 | 44.0 | 60.0 | 50.0 | 58.0 | 58.0 | 50.0 | 51.4 |

exhibits clear improvement over the base model in nearly all the studied programming languages. It is noteworthy that MCODER, despite being trained with very limited multilingual data, still outperforms other large language models (LLMs) of similar or even larger sizes.

**Multilingual Code Explanation.** Table 5 displays the Pass@1 results for multilingual code explanation tasks. The results show that GPT models still significantly outperform open-source models in the code explanation task. For markup languages (Json and Markdown), the complexity of the code structure makes it difficult to describe accurately in natural language, resulting in generally poorer performance. Code LLMs need instruction-following capabilities for such complex structures.

**Multilingual Code Completion.** The completion tasks consist of *single-line completion*, *multi-line completion*, *span completion*, and *span completion (light)*. As shown in Table 4, the Pass@1 results for multilingual code completion tasks indicate that GPT-4 Turbo still achieves the best performance. Additionally, since this task is relatively easier compared to code generation, some open-source models perform comparably to GPT-4 Turbo in certain programming languages.

## 5 FURTHER ANALYSIS

**Programming Classification.** In Figure 5, we categorize the programming languages of MCEVAL into 5 programming paradigms and 11 application scenarios and summarize the performance of code LLMs on the code generation task in Figure 6. It can be observed that code LLMs generally perform better in object-oriented and multi-paradigm programming languages (high-resource languages), while perform worse in functional and procedural programming languages (low-resource Languages). In areas like web development and scientific computing, the gap between open-source and closed-source models is narrowing. However, for application scenarios, there is still a substantial gap between open-source models and the closed-source GPT-4 series in low-resource languages related to scripting, mobile development, and educational research. MCODER performs superior over multiple same-size models and even some larger open-source models.

**Unbalance on Different Languages.** We compare the results of several open-source models on the MultiPL-E multilingual benchmark with corresponding languages on MCEVAL. We obtained scores for 11 programming languages (including Python, Java, JavaScript, C++, PHP, Rust, Swift, R, Lua, Racket, Julia) from the BigCode leaderboard.[1] As shown in Figure 7(1), due to the simplicity of Python language tasks in this dataset, many models exhibit significant score discrepancies between the two benchmarks. Figure 7(2) highlights a majority of models within the blue circle, indicating that the current state-of-the-art performance of most models primarily lies in high-resource languages like Python, while their proficiency in low-resource languages awaits further exploration and enhancement. By examining Figure 7(2) and (3), it becomes evident that all LLMs demonstrate consistent multilingual capabilities between MultiPL-E and MCEVAL.

---

[1] https://huggingface.co/spaces/bigcode/bigcode-models-leaderboard

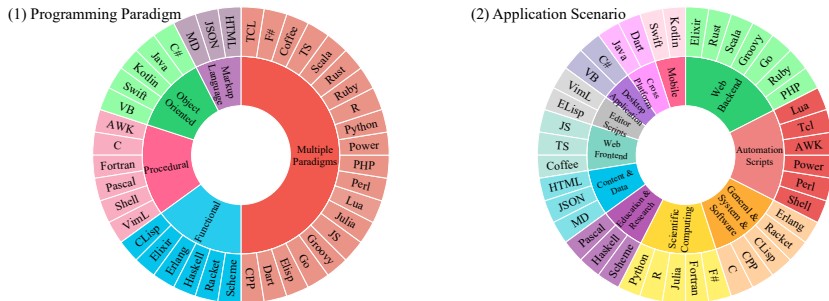

Figure 5: Classification of MCEVAL. The programming languages in MCEVAL can be categorized into 5 programming paradigms and 11 application scenarios.

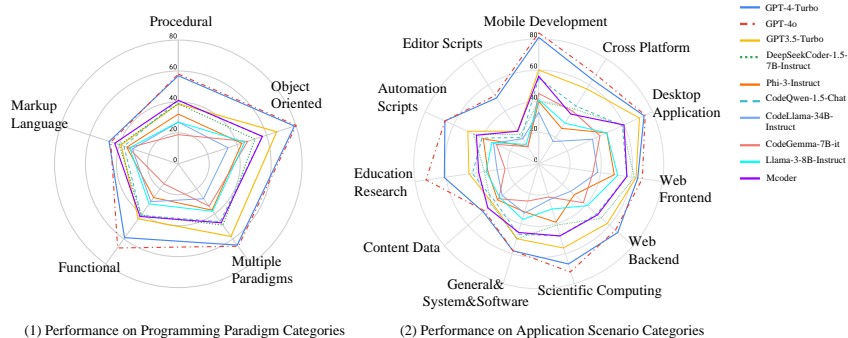

Figure 6: The performance of models in code completion tasks under different categories.

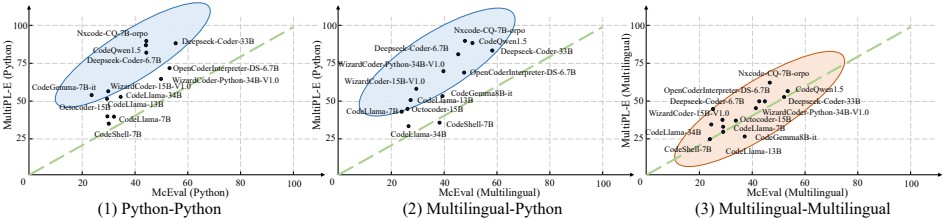

Figure 7: Unbalanced performance on different languages across MultiPL-E and our MCEVAL.

**Cross-lingual Transfer.** We fine-tune the CodeQwen-1.5 model using Python-only data in MCEVAL-INSTRUCT and compare it with MCODER. In Figure 8, CodeQwen-1.5 performs well in most high-resource languages, but CodeQwen without alignment exhibits unsatisfactory results in some low-resource languages due to the inability to follow instructions. As such, with fine-tuning using only Python data, CodeQwen-1.5-Python improves significantly across most languages. It shows that the CodeQwen foundation model already possesses strong coding capabilities but lacks adequate instruction-following skills. Therefore, fine-tuning with Python-only data can still effectively transfer instruction-following abilities to other languages, resulting in superior multilingual performance.

**Difficulty of MCEVAL.** Based on algorithmic complexity, we classify MCEVAL into three levels (Easy/Medium/Hard). In Figure 9, we conduct a statistical analysis of CodeQwen-1.5-Chat's performance on code generation tasks across various languages. For most languages, the code LLM can answer the majority of easy questions but struggles with medium and hard ones.

# 6   RELATED WORK

For the field of soft engineering, code LLMs (Feng et al., 2020; Chen et al., 2021; Scao et al., 2022; Li et al., 2022; Allal et al., 2023; Fried et al., 2022; Wang et al., 2021; Zheng et al., 2024; Guo et al., 2024) pre-trained on billions of code snippets, such as StarCoder (Li et al., 2023; Lozhkov et al.,

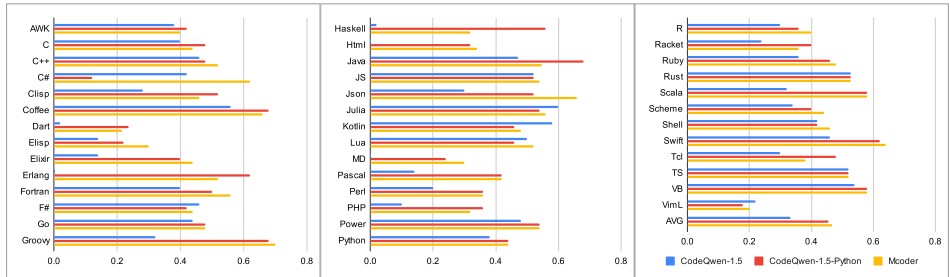

Figure 8: Cross-lingual transferability of LLMs among different languages. We fine-tune CodeQwen-1.5 using Python data in MCEVAL-INSTRUCT and OSS-Instruct to create CodeQwen-1.5-Python.

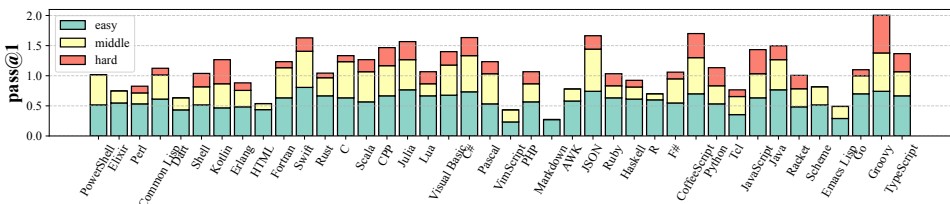

Figure 9: CodeQwen-1.5-Chat performance on MCEVAL for problems of different difficulty levels.

2024), CodeLlama (Rozière et al., 2023), DeepSeekCoder (Guo et al., 2024), and Code-Qwen (Hui et al., 2024). The development and refinement of code LLMs have been pivotal in automating software development tasks, and supporting code generation/translation/summarization.

Many benchmarks (Yu et al., 2024; Yin et al., 2023; Khan et al., 2023; Orlanski et al., 2023; Jain et al., 2024) have been woven to accurately assess code quality, functionality, and efficiency, such as HumanEval (Chen et al., 2021), MBPP (Austin et al., 2021), their upgraded version EvalPlus (Liu et al., 2023b). Studies have explored a variety of approaches, ranging from static evaluation using text matching to dynamic methods that involve code execution under a controlled environment. The current benchmarks support code LLMs to evaluate a series of different types of tasks, such as code understanding, function calling Zhuo et al. (2024), code repair (Lin et al., 2017; Tian et al., 2024; Jimenez et al., 2023; Zhang et al., 2023; Prenner & Robbes, 2023; He et al., 2022), code translation (Yan et al., 2023). Some works focus on the multilingual scenarios (Wang et al., 2023; Athiwaratkun et al., 2023; Peng et al., 2024; Zheng et al., 2023b) by extending the Python-only HumanEval/MBPP benchmark (e.g. MultiPL-E (Cassano et al., 2023)), which is challenged by the number of the languages.

## 7 CONCLUSION

In this work, we push a significant advancement in the assessment of code LLMs by proposing the first massively multilingual code evaluation benchmark (MCEVAL) by involving an annotation and verification process conducted by professional developers, which spans 40 programming languages and helps comprehensively tackle various tasks, including code generation, explanation, and completion. The multilingual SFT on created instruction corpora MCEVAL-INSTRUCT further emphasizes the proficiency of LLMs in multiple coding languages. Systematic evaluations of existing code LLMs on MCEVAL illuminate the performance disparities among open-source and closed-source models. Extensive multilingual multitask assessment on MCEVAL provides a realistic and comprehensive measurement of code LLMs, marking a leap forward for developers utilizing AI techniques to understand and generate code effectively across a wide spectrum of programming languages.

ACKNOWLEDGMENTS

This work was supported in part by the National Natural Science Foundation of China (Grant Nos. 62276017, 62406033, U1636211, 61672081), and the State Key Laboratory of Complex & Critical Software Environment (Grant No. SKLCCSE-2024ZX-18).

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

# A  APPENDIX

## A.1  LIMITATIONS

**Cover more languages and tasks**    Our work focuses on evaluating models on multilingual programming tasks, currently supporting assessments in 40 programming languages. Many models claim to support over 80 programming languages, so our work can continue to expand the range of programming languages, the number of test cases, and types of tasks to provide a more comprehensive evaluation of the models.

**Research on cross-lingual transfer**    We do not delve deeply into the cross-lingual transfer capabilities of models, leaving room for exploring a wider variety of models and models of different sizes. In the future, we will explore the programming language transferability scaling law of the code LLMs with different sizes.

## A.2  DATA ANNOTATION

### A.2.1  HUMAN ANNOTATION

To create the massively multilingual code evaluation benchmark MCEVAL, the annotation of multilingual code samples is conducted utilizing a comprehensive and systematic human annotation procedure, underpinned by rigorously defined guidelines to ensure accuracy and consistency.

We recruited annotators with backgrounds in computer software development from universities. During recruitment, we evaluated their programming and system operation skills to ensure they could handle tasks like question writing, code static analysis, and unit testing. Each annotator was assigned tasks based on their proficiency in specific programming languages.

Initially, 10 software developers in computer science are recruited as multilingual programming annotators with proven proficiency in the respective programming languages.

Following a detailed training session on the annotation protocol, annotators are tasked with creating problem definitions and the corresponding solution.

The guidelines for our annotation training session primarily cover the following aspects:

- **Standardized Format:** We provide an annotation example for 40 programming languages. Annotators are required to adhere to this standardized format when annotating data.

- **Accessibility:** The reference data for our annotations is sourced from materials available under permissive licenses, allowing unrestricted use and redistribution for research purposes.

- **Difficulty Level:** We provide annotators with detailed guidelines on the difficulty classification for each language. Annotators must strictly follow these guidelines to label problems according to their respective difficulty levels based on algorithmic complexity and functionality.

- **Self-Contained:** Annotators are required to thoroughly review their annotated problems to ensure that the problem descriptions include all necessary information for solving them without ambiguity. The provided example inputs and outputs must be correct, the reference answers must execute correctly, and the test cases written should comprehensively evaluate the accuracy of the functions.

### A.2.2  REFERENCE DATA

We use reference data (The annotators only draw the inspiration from the reference website, and create the question manually.) to draft questions and solutions with the help of GPT-4-Turbo. The draft questions initially may contain numerous issues, such as self-contained errors, inconsistent difficulty levels, overly simple unit tests, and incorrect unit tests. To address these issues, annotators revised and tested the draft questions to ensure that all questions and code were accurate and that the unit tests would pass. These reference data came from the following websites: For the algorithmic types of questions, we refer to the following websites:

- https://www.dotcpp.com/
- https://www.luogu.com.cn
- https://www.codecademy.com/
- https://www.codewars.com/

For markup language and design-type questions, we refer to the following websites:

- https://www.runoob.com/
- https://www.w3schools.com/

### A.2.3 QUALITY CONTROL

We adopt a dual-pass system to ensure the quality of our benchmark MCEVAL. First, one annotator labels the code snippets and their corresponding unit tests. Then, another annotator independently reviews these annotations, verifying their correctness, code accuracy, and unit test integrity. Following this, three senior annotators evaluate the overall unit test pass rate of the annotated dataset. If the pass rate exceeds 90%, the senior annotators proceed to perform additional reviews and corrections. Otherwise, the data is returned to the annotators for refinement. Finally, the canonical solution (labeled by the annotator) passes all corresponding test cases to ensure the correctness of each created problem (100% pass rate). This rigorous process ensures the creation of a high-quality, multilingual programming benchmark that supports in-depth analysis and understanding of code examples across diverse programming languages.

### A.2.4 ANNOTATION COSTS

We paid all the annotators the equivalent of $6 per question and provided them with a comfortable working environment, free meals, and souvenirs. We also provided the computer equipment and GPT-4 interface required for labeling. We labeled about 2,000 questions in total and employed them to check the quality of the questions/answers, and the total cost was about $12,000 in US dollars. The annotators checked the derived tasks, including multilingual code explanation and code completion.

### A.3 DETAILED ABOUT MCEVAL-INSTRUCT

Here we describe in detail the specific methods and processes of code sampling and quality control in the process of constructing MCEVAL-INSTRUCT.

**Code Sampling**

- (1) We crawled a large amount of code data from GitHub and preprocessed the data according to the StarCoder processing flow. Then, we can get a high-quality code dataset.
- (2) For each programming language, we randomly sampled the data to build an original dataset containing 40 languages, where the instances of each language are sampled to the same number of samples.
- (3) In addition, in the refined stage, we further used GPT-4 to refine the code snippets to the clearer and more standardized code (the code with docstring, code comments, and clear variables) and verified the refined code snippet through static code analysis and unit testing methods to ensure the correctness and quality of the code.

**Quality Control**   To check the correctness of the generated code from the GPT-4, we use static analysis and unit testing to ensure the accuracy of generated data as much as possible:

- Code Correctness: We use static code analysis tools and Abstract Syntax Tree (AST) parsing to ensure the correctness of the code.
- Automated Testing: We leverage an LLM to automatically generate unit tests, filtering out code that fails to pass these tests.
- Harmful Content Filtering: We filter out code snippets containing harmful information using keyword-based filters.

## A.4 EXAMPLES IN MCEVAL

Figure 10 display three examples of multilingual generation.

In Figure 11, we show three examples of multilingual explanation.

In Figure 12, we display three examples of multilingual explanation. The three examples from left to right correspond to the span completion task, the single-line completion task, and the multi-line completion task.

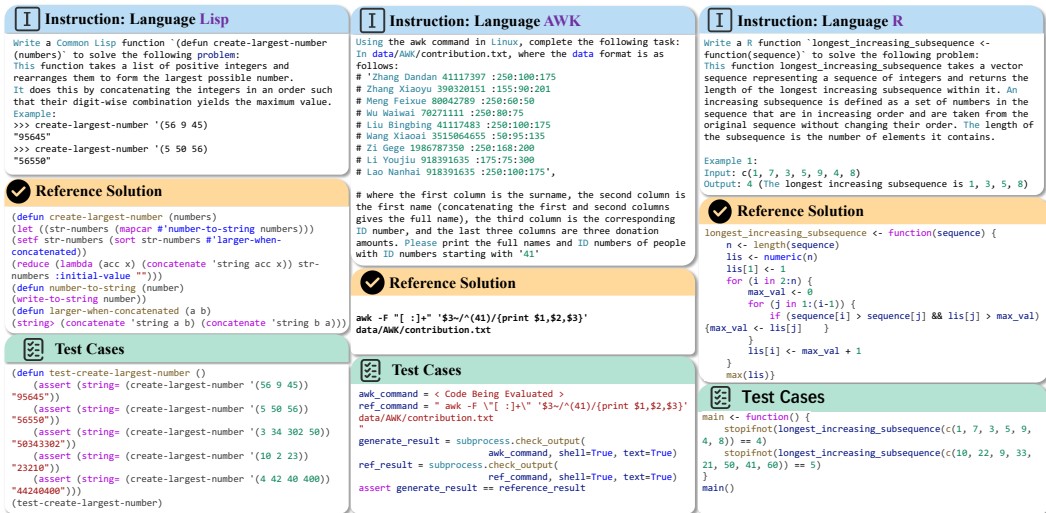

Figure 10: Examples of multilingual generation. The data mainly consists of an instruction part (including function name, function description, and function call cases), a reference solution, and a test cases part. **Left.** Shows an example of the Lisp language. **Middle.** Shows a file processing programming task in AWK language. During the evaluation, the corresponding file processing result by the generated code will be compared with the reference answer. **Right.** Shows an example of the R language.

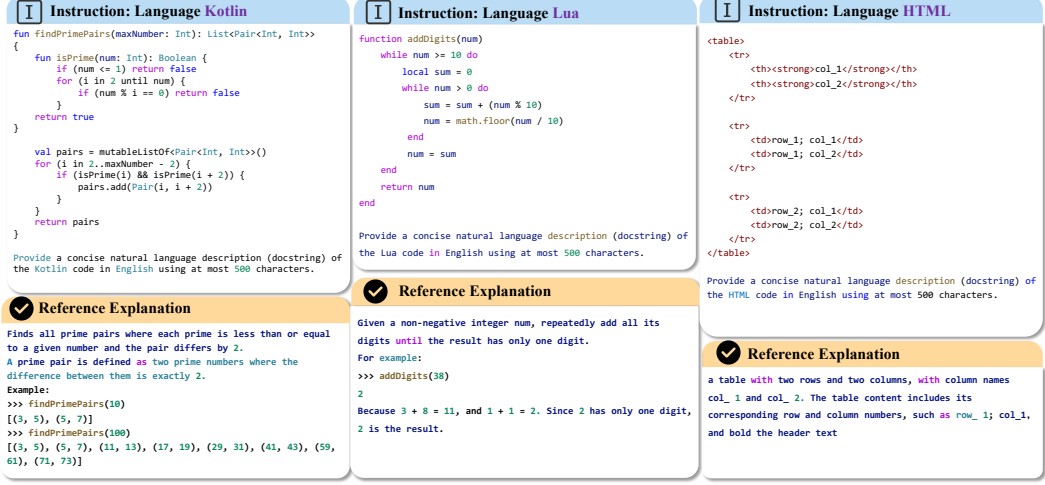

Figure 11: Examples of multilingual explanation. The data mainly consists of an instruction part (including a complete function), a reference Explanation. **Left.** Shows an example of the Kotlin language. **Middle.** Shows an example of the Lua language. **Right.** Shows an example of the HTML language.

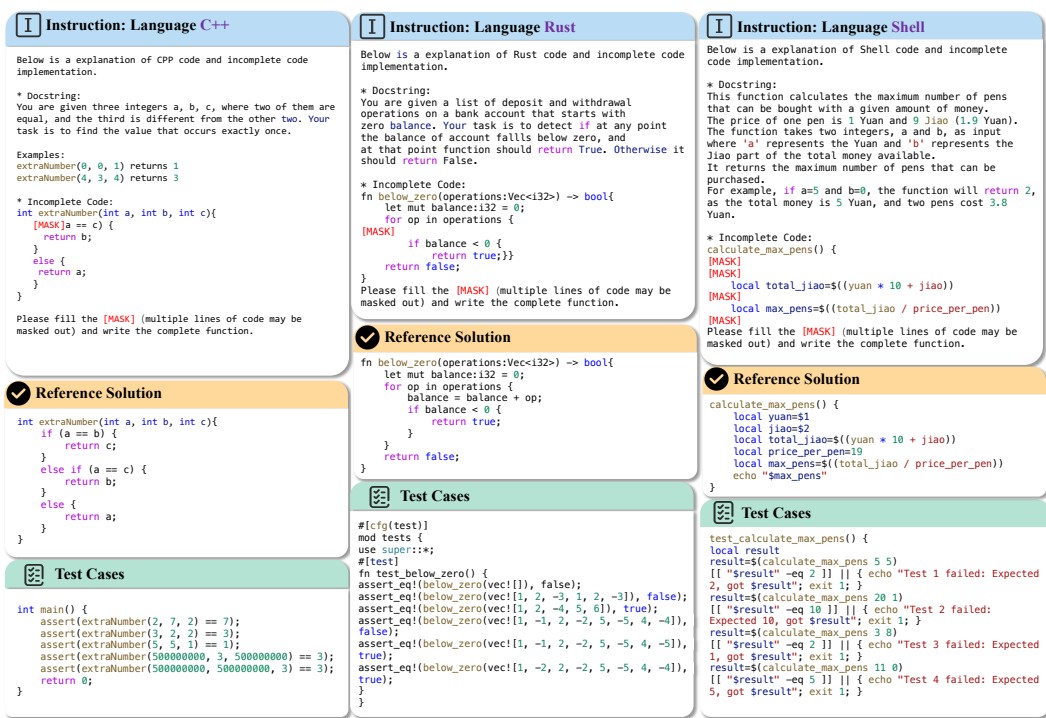

Figure 12: Examples of multilingual completion. The data mainly consists of an instruction part (including a incomplete function ), a reference complete code solution and test cases. **Left.** Shows an span completion example of the C++ language. **Middle.** Shows an single line completion example of the Rust language. **Right.** Shows an multiple line completion example of the Shell language.

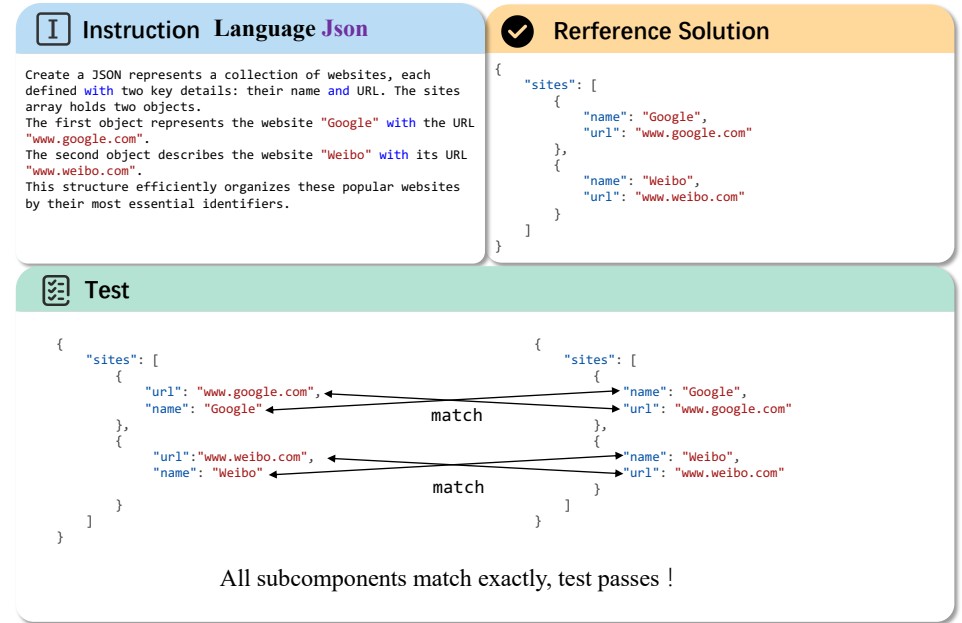

Figure 13: Examples of Markup language (Json) generation task evaluation.

## A.5   EVALUATION

For programming languages other than markup languages, we use an execution-based correctness metric by running the code with the provided test cases. For markup languages, we use the Exact

Table 6: Runtime environments for different programming languages.

| Language | Runtime Environments |
| --- | --- |
| AWK | GNU bash, version 4.4.20(1)-release (x86_64-pc-linux-gnu) |
| C | gcc (Ubuntu 7.5.0-3ubuntu1Ĩ8.04) 7.5.0 |
| C# | dotnet 8.0.100 |
| CPP | g++ (Ubuntu 7.5.0-3ubuntu1Ĩ8.04) 7.5.0 |
| CoffeeScript | CoffeeScript version 1.12.7 |
| Common Lisp | SBCL 1.4.5.debian |
| Dart | Dart SDK version: 3.3.1 (stable) |
| Elixir | elixir 1.3.3 |
| Emacs Lisp | GNU Emacs 25.2.2 |
| Erlang | Erlang/OTP 20 [erts-9.2] |
| F# | dotnet 8.0.100 |
| Fortran | GNU Fortran (Ubuntu 7.5.0-3ubuntu1Ĩ8.04) 7.5.0 |
| Go | go version go1.18.4 linux/amd64 |
| Groovy | Groovy Version: 4.0.16 JVM: 17.0.9 Vendor: Oracle Corporation OS: Linux |
| HTML | - |
| Haskell | The Glorious Glasgow Haskell Compilation System, version 9.4.7 |
| Json | - |
| Java | javac 11.0.19 |
| JavaScript | Node.js v16.14.0 |
| Julia | julia v1.9.4 |
| Kotlin | kotlinc-jvm 1.9.21 (JRE 17.0.9+11-LTS-201) |
| Lua | Lua 5.4.6 Copyright (C) 1994-2023 Lua.org, PUC-Rio |
| Markdown | - |
| PHP | PHP 7.2.24-0ubuntu0.18.04.17 (cli) (built: Feb 23 2023 13:29:25) ( NTS ) |
| Pascal | Free Pascal Compiler version 3.2.2 [2021/05/16] for x86_64 |
| Perl | perl 5, version 26, subversion 1 (v5.26.1) built for x86_64-linux-gnu-thread-multi |
| PowerShell | PowerShell 7.4.0 |
| Python | Python 3.8.12 |
| R | R version 3.4.4 |
| Racket | Racket v6.11 |
| Ruby | ruby 2.5.1p57 (2018-03-29 revision 63029) [x86_64-linux-gnu] |
| Rust | rustc 1.74.0 (79e9716c9 2023-11-13) |
| Scala | Scala code runner version 3.3.1 – Copyright 2002-2023, LAMP/EPFL |
| Scheme | Racket v6.11 |
| Shell | GNU bash, version 4.4.20(1)-release (x86_64-pc-linux-gnu) |
| Swift | Swift version 5.9.2 (swift-5.9.2-RELEASE) |
| Tcl | tclsh 8.6.11 |
| TypeScript | tsc Version 5.3.3 |
| VimScript | VIM - Vi IMproved 9.0 (2022 Jun 28, compiled Dec 20 2023 18:57:50) |
| Visual Basic | dotnet 8.0.100 |

Match metric for evaluation. Taking Json as an example, we parse all subcomponents in Json. If the model result is exactly the same as the subcomponent of the reference solution, the model generation result is considered correct. An example of Markup language (Json) is shown in Figure 13.

We adopt the greedy Pass@1 (%) metric (Kulal et al., 2019; Chen et al., 2021) for our evaluations. For closed-source models, we generate answers through the official API service. For open-source models, we prioritize using vLLM (Kwon et al., 2023) for faster inference if the model is supported by vLLM. Otherwise, we perform inference with the Distributed Data Parallel (DDP) module from PyTorch. For the code generation and code completion tasks, we extract the functional part of the code from the model outputs and combine it with corresponding test cases to form compilable and executable code. For the code explanation task, we adopt a two-pass generation approach (Code-to-Natural-Language and Natural-Language-to-Code). The extraction and execution process for this task is consistent with the previous two tasks. We conduct all evaluations in a Docker environment. Detailed information on the code compilation and execution environment are displayed in Table 6. We have uploaded the Docker image to docker hub to facilitate the reproduction of results and the evaluation of new models.

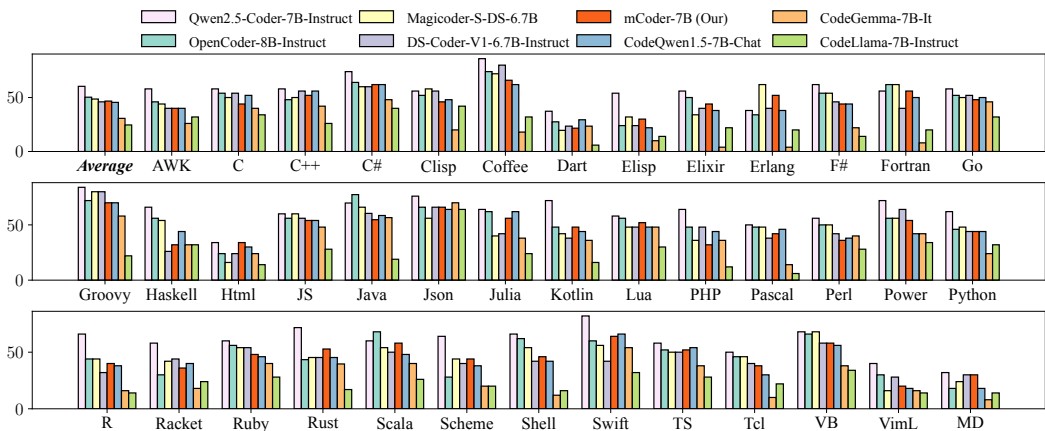

Figure 14: Pass@1 (%) scores of different code LLMs (<10B) for multilingual code generation tasks on MCEVAL. "AVG" represents the average scores of all code languages.

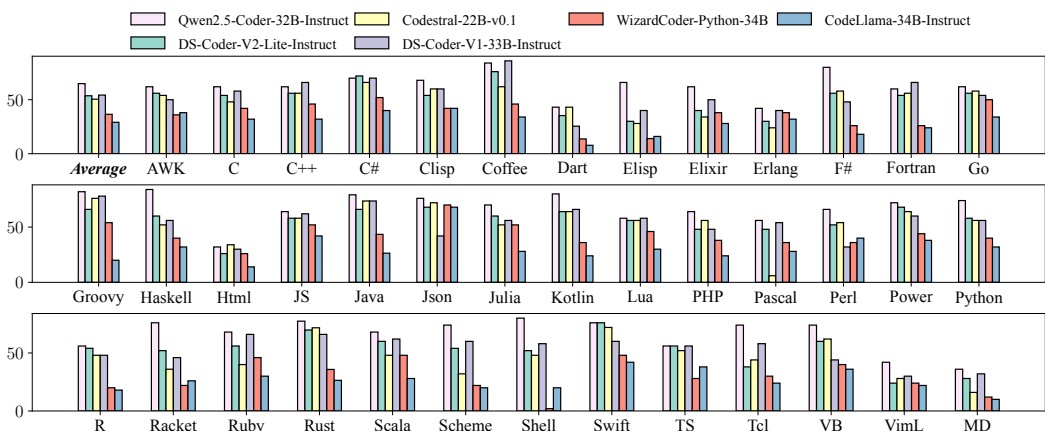

Figure 15: Pass@1 (%) scores of different code LLMs (10B to 40B) for multilingual code generation tasks on MCEVAL. "AVG" represents the average scores of all code languages.

## A.6 OPTIMIZATION DETAILS

All MCODER models are fine-tuned using 8 NVIDIA A800-80GB GPUs. The models are trained for 2 epochs with a cosine scheduler, starting at a learning rate of 2e-5 and incorporating a 3% warmup phase. Training a model takes about 5 hours. We used AdamW (Loshchilov & Hutter, 2017) as the optimizer and a batch size of 512 with a sequence truncation length of 4096. We use PyTorch's Fully Sharded Data Parallel (FSDP) to perform distributed training of the model, and use gradient checkpointing technology and gradient accumulation to save memory and achieve training with a larger batch size.

## A.7 EXTRA RESULTS

## A.8 PROGRAMMING CLASSIFICATION

As shown in Table 7 and Table 8, we comprehensively display the code generation performance of the models we tested across various programming paradigms and application scenarios.

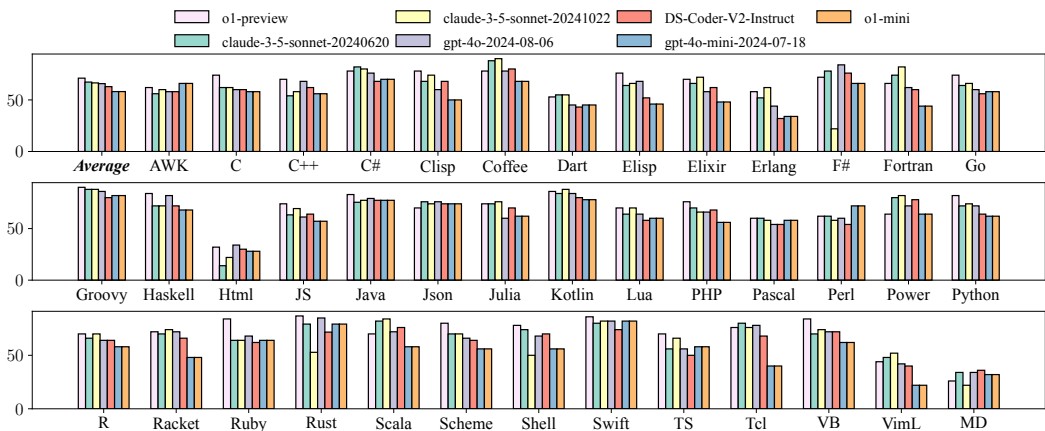

Figure 16: Pass@1 (%) scores of different code LLMs (Closed Source & 200B+) for multilingual code generation tasks on MCEVAL. "AVG" represents the average scores of all code languages.

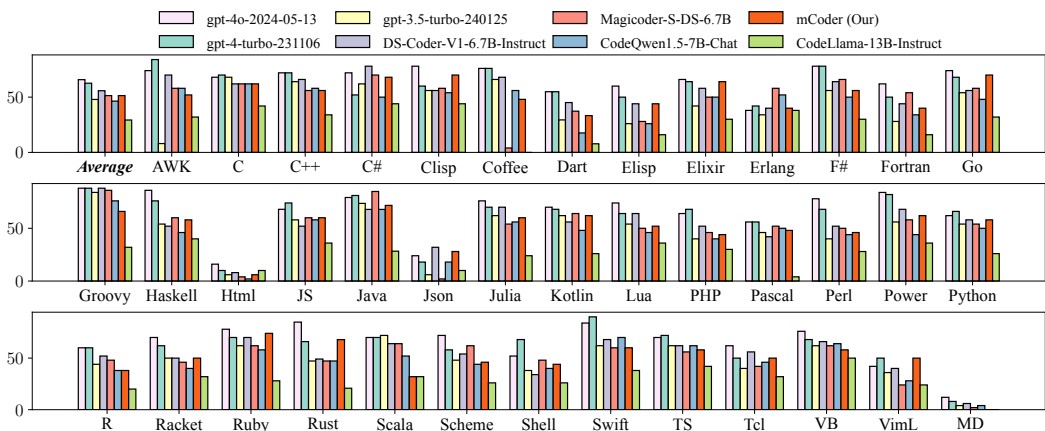

Figure 17: Pass@1 (%) scores of different code LLMs for multilingual code explain tasks on MCEVAL. "AVG" represents the average scores of all code languages.

## A.9 MCODER RESULT

In Table 9, we show some extra MCODER Pass@1 (%) results on multilingual code generation tasks. We evaluate the base models CodeQwen-1.5 and DeepsSeek-Coder-1.5 respectively. In addition to CodeQwen-1.5, we also selected DeepSeek-Coder-1.5-base as the base model for fine-tuning.

## A.10 PARALLEL QUESTIONS ACROSS LANGUAGES & PROGRAMMING GRAMMAR

Due to the large number of languages, it is difficult to ensure parallel problem annotation. For most language annotations, we follow the characteristics of the language and perform independent annotations. For example, structured languages such as Markdown and HTML need independent annotations. For some similar languages, such as Typescript and Javascript, we use parallel annotation on some data.

As shown in Figure 19, we analyzed the programming languages in the MCEVAL from the representation perspective. We used CodeBERT (Feng et al., 2020) to extract code representations from code snippets in MCEVAL. These representations were visualized using t-SNE (Van der Maaten & Hinton, 2008) and hierarchical clustering (Murtagh & Contreras, 2012) methods. The figure clearly shows that languages with similar syntax have closely related representations. For example, other functional programming languages similar to Common Lisp, as well as C, C++, Java, and scripting languages, exhibit high grammar similarity.

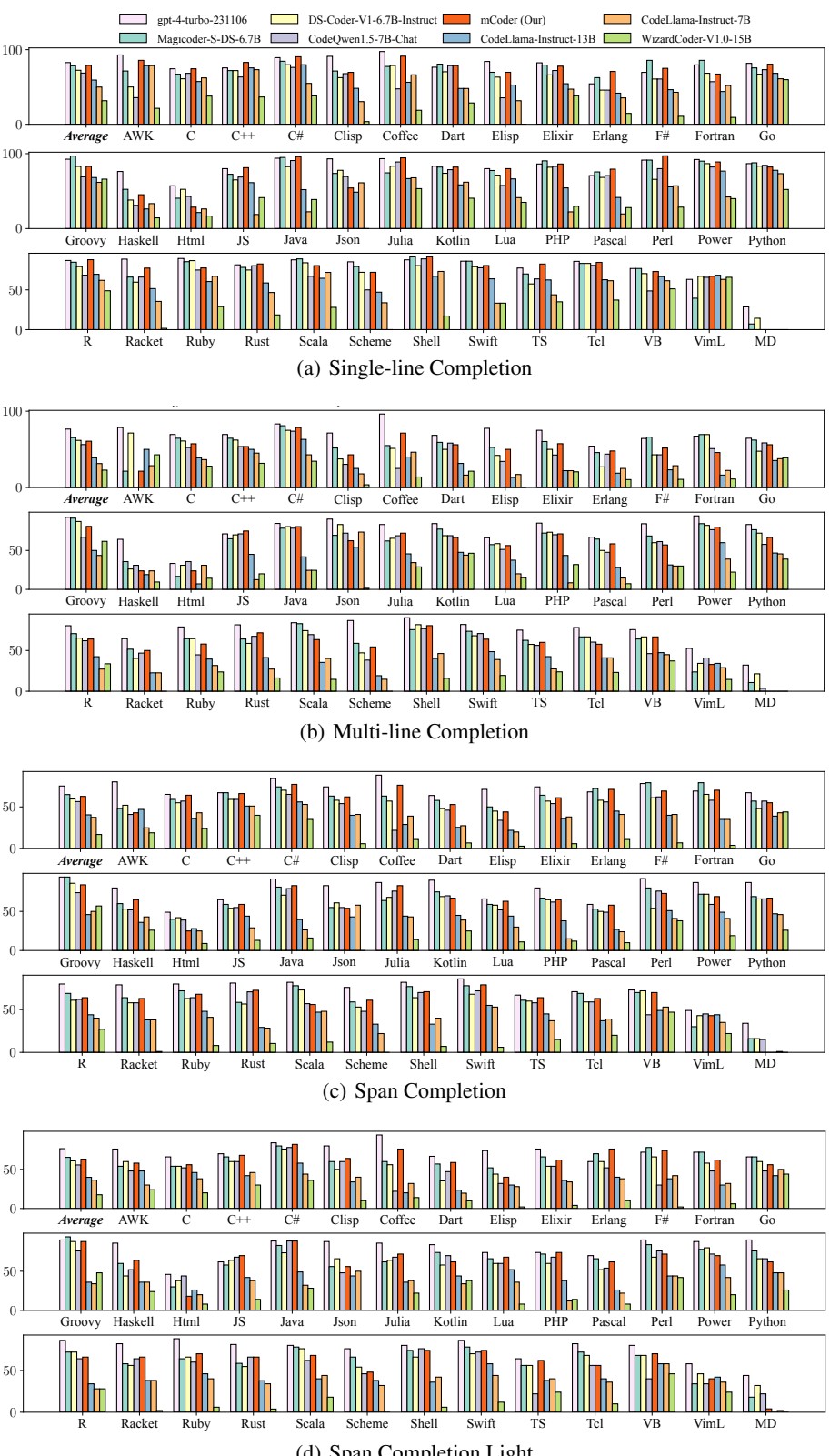

Figure 18: Pass@1 (%) scores of different models for multilingual code completion tasks on MCEVAL. "Avg" represents the average scores of all code languages.

Table 7: Pass@1(%) results of code generation performance of across various programming paradigms

| Method | Procedural | Object Oriented | Multiple Paradigms | Functional | Markup Language |
|---|---|---|---|---|---|
| GPT-4o (240517) | **58.0** | **79.8** | **65.9** | **67.0** | 46.0 |
| GPT-4 Turbo (231106) | 56.7 | 78.7 | 65.2 | 59.3 | **46.7** |
| GPT-3.5-Turbo (240125) | 38.7 | 66.8 | 57.6 | 44.3 | 39.3 |
| Codegemma-7b-it | 19.3 | 46.6 | 34.0 | 16.3 | 34.0 |
| CodeLlama-13b-Instruct | 21.3 | 32.0 | 27.0 | 32.3 | 28.0 |
| CodeLlama-34b-Instruct | 27.3 | 33.6 | 28.0 | 30.0 | 30.7 |
| CodeLlama-7b | 20.3 | 28.1 | 23.4 | 26.7 | 30.7 |
| CodeQwen-1.5-7b-Chat | 41.3 | 57.3 | 46.3 | 41.0 | 37.3 |
| Codeshell-7b-chat | 16.0 | 24.1 | 25.7 | 14.0 | 34.7 |
| Codestral-22B-v0.1 | 40.0 | 67.6 | 54.1 | 39.7 | 40.7 |
| DeepSeekCoder-33b-instruct | 52.7 | 62.8 | 56.3 | 52.0 | 34.7 |
| DeepSeekCoder-1.5-7b-instruct | 39.0 | 51.8 | 48.8 | 41.0 | 40.0 |
| Magicoder-S-DS-6.7B | 45.7 | 58.5 | 49.4 | 49.0 | 32.0 |
| Llama-3-8B-Instruct | 27.3 | 44.7 | 38.0 | 32.0 | 33.3 |
| Nxcode-CQ-7B-orpo | 40.7 | 54.9 | 45.5 | 41.3 | 36.7 |
| OCTOCODER | 20.7 | 28.9 | 21.9 | 25.0 | 25.3 |
| OpenCodeInterpreter-DS-6.7B | 40.7 | 57.7 | 46.4 | 42.0 | 42.0 |
| Phi-3-medium-4k-instruct | 32.3 | 43.1 | 36.6 | 26.7 | 35.3 |
| Qwen1.5-72B-Chat | 38.3 | 37.2 | 36.2 | 29.3 | 39.3 |
| WizardCoder-15B-V1.0 | 19.0 | 31.6 | 34.2 | 24.0 | 6.7 |
| WizardCoder-Python-34B | 27.7 | 43.9 | 38.2 | 33.7 | 36.0 |
| MCODER | 41.3 | 57.3 | 47.4 | 42.3 | 43.3 |

Table 8: Pass@1(%) results of code generation performance of across various application scenarios

| Method | Mobile | Cross | Desktop | Frontend | Backend | Scientific | General | Content | Education | Scripts | Editor |
|---|---|---|---|---|---|---|---|---|---|---|---|
| GPT-4o (230517) | **84.0** | **68.3** | **75.0** | **66.7** | 64.6 | **71.6** | **57.6** | 46.0 | **72.7** | 65.7 | **52.0** |
| GPT-4 Turbo (231106) | 81.0 | 64.4 | 74.0 | 64.0 | **66.6** | 66.8 | **57.6** | **46.7** | 60.7 | **65.7** | 50.0 |
| GPT-3.5 (240125) | 60.0 | 56.7 | 71.0 | 63.3 | 57.5 | 55.6 | 50.4 | 39.3 | 45.3 | 50.0 | 25.0 |
| Codegemma-7b-it | 45.0 | 40.4 | 43.0 | 34.7 | 37.7 | 21.6 | 24.8 | 34.0 | 22.0 | 29.7 | 13.0 |
| code-Llama-13b | 30.0 | 15.4 | 39.0 | 34.7 | 28.0 | 23.2 | 34.8 | 28.0 | 27.7 | 13.0 | 13.0 |
| CodeLlama-34b-Instruct | 33.0 | 17.3 | 38.0 | 38.0 | 27.2 | 24.0 | 32.8 | 30.7 | 26.7 | 31.7 | 19.0 |
| Code-Llama-7b-Instruct | 24.0 | 12.5 | 37.0 | 29.3 | 22.7 | 20.8 | 29.2 | 30.7 | 19.3 | 27.0 | 14.0 |
| CodeQwen-1.5-7b | 55.0 | 44.2 | 59.0 | 56.7 | 48.7 | 47.6 | 46.8 | 37.3 | 42.7 | 40.0 | 20.0 |
| Codeshell-7b-chat | 23.0 | 14.4 | 26.0 | 40.7 | 26.1 | 17.2 | 21.2 | 34.7 | 13.3 | 22.7 | 8.0 |
| Codestral-22B-v0.1 | 68.0 | 58.7 | 64.0 | 57.3 | 55.0 | 54.0 | 44.8 | 40.7 | 30.0 | 53.3 | 28.0 |
| DeepSeekCoder-33b-instruct | 63.0 | 50.0 | 57.0 | 68.0 | 60.6 | 54.8 | 54.0 | 34.7 | 56.7 | 52.7 | 35.0 |
| DeepSeekCoder-1.5-7b-instruct | 40.0 | 42.3 | 59.0 | 62.0 | 52.7 | 40.8 | 50.0 | 40.0 | 34.7 | 46.0 | 22.0 |
| Magicoder-S-DS-6.7B | 49.0 | 43.3 | 64.0 | 60.7 | 50.4 | 49.6 | 52.4 | 32.0 | 48.7 | 49.7 | 24.0 |
| Llama-3-8B-Instruct | 41.0 | 30.8 | 48.0 | 50.7 | 40.5 | 30.0 | 37.2 | 33.3 | 34.0 | 33.0 | 15.0 |
| Nxcode-CQ-7B-orpo | 54.0 | 40.4 | 55.0 | 53.3 | 48.4 | 48.0 | 46.8 | 36.7 | 42.7 | 39.7 | 20.0 |
| OCTOCODER | 22.0 | 20.2 | 33.0 | 28.7 | 21.8 | 16.4 | 27.2 | 25.3 | 16.0 | 29.0 | 14.0 |
| OpenCodeInterpreter-DS-6.7B | 47.0 | 42.3 | 64.0 | 58.0 | 45.9 | 47.6 | 46.4 | 42.0 | 43.3 | 44.0 | 24.0 |
| Phi-3-medium-4k-instruct | 40.0 | 26.9 | 48.0 | 48.7 | 30.3 | 39.2 | 31.6 | 35.3 | 33.3 | 39.0 | 13.0 |
| Qwen1.5-72B-Chat | 30.0 | 29.8 | 43.0 | 44.7 | 36.3 | 30.4 | 38.0 | 39.3 | 32.7 | 40.0 | 21.0 |
| WizardCoder-15B-V1.0 | 28.0 | 24.0 | 36.0 | 48.0 | 37.1 | 29.2 | 27.2 | 6.7 | 20.7 | 26.3 | 9.0 |
| WizardCoder-Python-34B | 42.0 | 28.8 | 46.0 | 42.0 | 44.2 | 32.8 | 38.0 | 36.0 | 32.7 | 32.3 | 19.0 |
| MCODER | 56.0 | 38.5 | 60.0 | 57.3 | 50.4 | 48.0 | 46.0 | 43.3 | 39.3 | 44.3 | 25.0 |

We selected training data from several languages in MCEVAL-INSTRUCT, which exhibit significant grammatical differences (approximately 10K samples of Python and 1K samples for other languages) and fine-tuned the model. The results are as shown in Table 10.

When trained using **only Python data**, the performance on Python and AWK improved. However, this led to the scores for TypeScript and JavaScript dropping to 0. Upon inspection, we found that the generated code for these two languages contained syntax errors (Less data may lead to instability in model training).

When training on **a mixture of several languages**, Python performance decreased slightly compared to using only Python data, while Scheme performance improved significantly. Furthermore, the syntax generation for TypeScript and JavaScript returned to normal (even without adding JavaScript data, as TypeScript and JavaScript share similar syntax). However, there was no significant improvement compared to the base model.

Thus, fine-tuning multilingual code models presents significant challenges. Similar languages can provide mutual benefits, while languages with greater differences may negatively impact performance.

Table 9: Additional MCODER Pass@1 (%) results on multilingual code generation tasks. "Avg$_{all}$" represents the average Pass@1 scores across all programming languages in the MCEVAL. Here, MCODER-DS indicates that the fine-tuned base model is DeepSeekCoder-1.5-7b-base.

| Method | Size | AWK | C | C++ | C# | Clisp | Coffee | Dart | Elisp | Elixir | Erlang | Fortran | F# | Go | Groovy | Haskell | Html | Java | JS | Json | Julia |
|---|---|---|---|---|---|---|---|---|---|---|---|---|---|---|---|---|---|---|---|---|---|
| DeepSeekCoder-1.5-base | 7B | 30.0 | 36.0 | 38.0 | 40.0 | 40.0 | 58.0 | 0.0 | 18.0 | 2.0 | 14.0 | 50.0 | 44.0 | 48.0 | 26.0 | 2.0 | 4.0 | 49.1 | 32.0 | 16.0 | 34.0 |
| CodeQwen-1.5 | 7B | 38.0 | 40.0 | 46.0 | 42.0 | 28.0 | 56.0 | 2.0 | 14.0 | 14.0 | 0.0 | 40.0 | 46.0 | 44.0 | 32.0 | 2.0 | 0.0 | 47.2 | 52.0 | 30.0 | 60.0 |
| CodeQwen-1.5-Python | 7B | 42.0 | 48.0 | 48.0 | 12.0 | 52.0 | 68.0 | 23.5 | 22.0 | 40.0 | 62.0 | 50.0 | 42.0 | 48.0 | 68.0 | 56.0 | 32.0 | 67.9 | 52.0 | 52.0 | 54.0 |
| **MCODER-DS** | 7B | 34.0 | 46.0 | 50.0 | 26.0 | 30.0 | 72.0 | 19.6 | 6.0 | 26.0 | 24.0 | 58.0 | 30.0 | 48.0 | 12.0 | 26.0 | 28.0 | 67.9 | 48.0 | 62.0 | 48.0 |
| **MCODER** | 7B | 40.0 | 44.0 | 52.0 | 62.0 | 46.0 | 66.0 | 21.6 | 30.0 | 44.0 | 52.0 | 56.0 | 44.0 | 48.0 | 70.0 | 32.0 | 34.0 | 54.7 | 54.0 | 66.0 | 56.0 |

| Method | Kotlin | Lua | MD | Pascal | Perl | PHP | Power | Python | R | Racket | Ruby | Rust | Scala | Scheme | Shell | Swift | Tcl | TS | VB | VimL | Avg$_{all}$ |
|---|---|---|---|---|---|---|---|---|---|---|---|---|---|---|---|---|---|---|---|---|---|
| DeepSeekCoder-1.5-7B-base | 42.0 | 20.0 | 0.0 | 24.0 | 24.0 | 36.0 | 42.0 | 54.0 | 24.0 | 20.0 | 38.0 | 39.6 | 44.0 | 20.0 | 18.0 | 32.0 | 10.0 | 44.0 | 22.0 | 22.0 | 28.9 |
| CodeQwen-1.5 | 58.0 | 50.0 | 0.0 | 14.0 | 20.0 | 10.0 | 48.0 | 38.0 | 30.0 | 24.0 | 36.0 | 52.8 | 32.0 | 34.0 | 42.0 | 46.0 | 30.0 | 52.0 | 54.0 | 22.0 | 33.2 |
| CodeQwen-1.5-Python | 46.0 | 46.0 | 24.0 | 42.0 | 36.0 | 36.0 | 54.0 | 44.0 | 36.0 | 40.0 | 40.0 | 52.8 | 58.0 | 40.0 | 42.0 | 62.0 | 48.0 | 52.0 | 58.0 | 18.0 | 45.5 |
| **MCODER-DS** | 36.0 | 42.0 | 22.0 | 34.0 | 8.0 | 34.0 | 46.0 | 42.0 | 22.0 | 40.0 | 56.0 | 45.3 | 48.0 | 30.0 | 38.0 | 48.0 | 34.0 | 46.0 | 50.0 | 28.0 | 37.8 |
| **MCODER** | 48.0 | 52.0 | 30.0 | 42.0 | 36.0 | 32.0 | 54.0 | 44.0 | 40.0 | 36.0 | 48.0 | 52.8 | 58.0 | 44.0 | 46.0 | 64.0 | 38.0 | 52.0 | 58.0 | 20.0 | 46.7 |

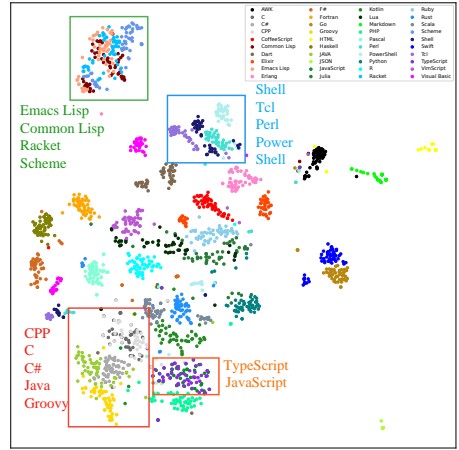

(1) Representation visualization based on t-SNE

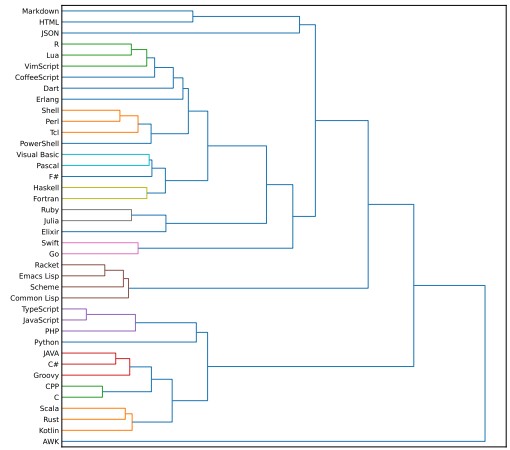

(2) Representation visualization based on Hierarchical Cluster

Figure 19: Analysis from the representation perspective on MCEVAL. Languages with similar syntax have closely related representations

Table 10: Preliminary explorations on the impact of finetuning across different languages on model performance.

| Setting | Python | Scheme | TypeScript | JavaScript | AWK |
|---|---|---|---|---|---|
| CodeQwen1.5-base | 38.0 | 34.0 | **52.0** | **52.0** | 38.0 |
| + Python | **48.0** | 12.0 | 0.0 | 0.0 | **40.0** |
| + Python&Scheme&TypeScript&AWK | 44.0 | **38.0** | 50.0 | 48.0 | **42.0** |

## A.11 DETAILED RELATED WORK

**Code Large Language Model.** In recent years, numerous large language models (LLMs) have been developed specifically for code-related tasks. For the field of soft engineering, code LLMs (Feng et al., 2020; Chen et al., 2021; Scao et al., 2022; Li et al., 2022; Allal et al., 2023; Fried et al., 2022; Wang et al., 2021; Zheng et al., 2024; Guo et al., 2024) pre-trained on billions of code snippets, such as StarCoder (Li et al., 2023; Lozhkov et al., 2024), CodeLlama (Rozière et al., 2023), DeepSeekCoder (Guo et al., 2024), and Code-Qwen (Bai et al., 2023). The development and refinement of code LLMs have been pivotal in automating software development tasks, providing code suggestions, and supporting code generation/translation.

To improve the performance of code generation, researchers used optimized prompts (Liu et al., 2023a; Reynolds & McDonell, 2021; Zan et al., 2023; Beurer-Kellner et al., 2023), bring test cases (Chen et al., 2023) and collaborative roles (Dong et al., 2023). There are also some related studies on using large language models for other code tasks, such as dynamic programming (Dagan et al., 2023), compiler optimization (Cummins et al., 2023), multi-lingual prompts (Di et al., 2023), and Program of Thoughts (Chen et al., 2022).

**Code Evaluation.** In the domain of code evaluation, a rich tapestry of benchmarks (Zheng et al., 2023b; Yu et al., 2024; Yin et al., 2023; Peng et al., 2024; Khan et al., 2023; Orlanski et al., 2023) has been woven to address the challenges of accurately assessing code quality, functionality, and efficiency, such as HumanEval (Chen et al., 2021), MBPP (Austin et al., 2021), their upgraded version EvalPlus (Liu et al., 2023b). Studies have explored a variety of approaches, ranging from static analysis techniques (e.g. exact match (EM) and edit similarity (ES)), which examine code without executing it, to dynamic methods that involve code execution in controlled environments (e.g. Pass@k). The current benchmarks support code models to evaluate a series of different types of tasks, such as code understanding, function calling (Zhuo et al., 2024), code repair (Lin et al., 2017; Tian et al., 2024; Jimenez et al., 2023; Zhang et al., 2023; Prenner & Robbes, 2023; He et al., 2022), code translation (Yan et al., 2023). Recently, many works Wei et al. (2023); Zhuo et al. (2024) have leveraged LLMs to construct large-scale evaluation datasets and instruction-tuning corpora, further enhancing the evaluation and performance of code models. In our work, we used a similar approach to construct an instruction dataset and proposed the Cross-lingual Code Transfer method to expand the number of languages to 40. Some recent works pay attention to the multilingual scenarios (Cassano et al., 2023; Wang et al., 2023; Athiwaratkun et al., 2023; Zheng et al., 2023a; Peng et al., 2024; Zheng et al., 2023b) by extending the existing python-only HumanEval or MBPP benchmark, such as MultiPL-E (Cassano et al., 2023) and MBXP (Athiwaratkun et al., 2023), which is challenged by the number of the covering languages and data leaking problem (Li et al., 2023; Jain et al., 2024).

