# OpenReview forum: "McEval: Massively Multilingual Code Evaluation"
_ICLR.cc/2025/Conference — ICLR 2025 Poster_

### Official Review · Reviewer_3QMs · 2024-10-24

**Soundness:** 3
**Presentation:** 2
**Contribution:** 2
**Rating:** 6
**Confidence:** 4

**Summary:**

This paper presents MCEval,  a massively multilingual code benchmark covering 40 programming languages with 16K test samples. MCEval provides a new evaluation benchmark in multilingual scenarios. In addition, the authors also present a new instruction dataset MCEVAL-INSTRUCT for multilingual code LLMs. The experiments results suggest that the benchmark MCEval could measure the gap between open-source and closed-source models.

**Strengths:**

1. This paper is easy to follow. The figures presented are clear and easy to understand.
2. The benchmark MCEval is comprehensive, covering 40 programming languages with 16K test samples.

**Weaknesses:**

1. The contribution of MCoder is unclear.
The contribution of this paper is comprise of two parts, a multilingual evaluation benchmark MCEval and a multilingual code LLM MCoder (trained on an instruction corpora MCEVAL-INSTRUCT). However, the performance of MCoder (using gpt-4 to generate data) are still behind the previous sota code LLMs like magicoder who uses gpt3.5 to generate instruction data.  The authors claim that MCoder is used as a strong baseline for MCEVAL(line 073), but why others don't use a better model (Magicoder) but MCoder as their baseline?

2. The method of constructing dataset is lack of novelty.
The biggest contribution of this paper seems the number of programming languages included in their dataset. However, the method of constructing such their dataset is using LLM to generate and refine from the code snippets, which has been used in many previous works such as bigcodebench(https://arxiv.org/pdf/2406.15877).

**Questions:**

1. What impact will the Unbalance on Different Languages (in section 5) bring to the evaluation process of code LLMs？
2. What kinds of languages will help code LLMs to generate better? What will do harm?  These problems maybe more insightful for multilingual code evaluation of coding LLMs.

---

> ### Author Response · Authors · 2024-11-18
> **Authors' Response (1/2)**
>
> > **Q1**: The contribution of MCoder is unclear.
>
> Sorry for any confusion. We believe that Magicoder is an excellent choice for application implementation!
> In our work, the contribution of MCoder mainly includes two aspects: On the one hand, it **verifies the effectiveness of the multilingual instruction corpora McEval-Instruct** proposed in our section 3.1, that is, it can be used to improve the multilingual coding capability of code models. On the other hand, we also conducted **some insight multilingual transfer exploration experiments** through MCoder. In the *Cross-lingual Transfer section of Section 5*, as shown in Figure 5, we finetune the CodeQwen-1.5 model using Python-only data in MCEVAL-INSTRUCT and compare it with the MCoder. As such, with finetuning using only Python data, CodeQwen-1.5 improves significantly across most languages.
>
> We sincerely hope that our corpus and exploratory experiments can bring some help to future multilingual programming research!
>
> > **Q2**: The method of constructing such dataset has been used in many previous works such as bigcodebench.
>
> This is a critical concern, and we apologize for not providing sufficient detail in the main text.
>
> BigCodeBench is a very extraordinary work! As far as we know, this work was proposed in June this year, and our work was also completed in June. Thank you for your reminder. We have updated the related work section of the paper to include a description of BigCodeBench!
> In addition, some works such as BigcodeBench mainly focus on Python language, while our work focuses on multilingual programming(data annotation, testing sandbox construction, and code snippet extraction method across 40 languages). This method of constructing datasets is commonly adopted by many works. We believe that it is scientific and effective, and can provide more reliable and higher-quality datasets.
>
> This is also our goal, to provide large-scale high-quality multilingual code evaluation datasets for the code open-source community. We hope our work, along with outstanding projects like BigCodeBench and MagiCoder, can help drive advancements in the field of code!

---

> ### Author Response · Authors · 2024-11-18
> **Authors' Response (2/2)**
>
> > **Q3**: What impact will the Unbalance on Different Languages bring to the evaluation process of code LLMs？
>
> Thank you for the excellent question.
> In our experiments, as shown in Figure 7(2), many models that perform exceptionally well on Multil-E Python exhibit lower performance on McEval, highlighting the uneven performance of numerous large code models across different programming languages. We believe that multilingual capabilities in code are essential, and evaluating programming abilities should not be limited to popular languages like Python. Many code LLMs claim support for over 80 languages, so more comprehensive datasets are needed to validate and compare these capabilities. In our dataset, we maintain over 50 examples per language to ensure quantitative balance. However, our evaluation shows that many models perform poorly, even scoring zero in low-resource languages. We hope our dataset can support the development of more robust code models through ongoing iteration.
>
> > **Q4**: What kinds of languages will help code LLMs to generate better? What will do harm?
>
> That's a profound insight! We sincerely hope that our dataset can contribute to the exploration of these insightful questions.
> Based on our exploration, we believe that high-quality data in a specific language can enhance the performance of that language and the syntax of similar languages. Meanwhile, it may harm the performance of languages with different syntaxes.
>
> In our work, we conducted preliminary explorations in two areas:
>
> 1. **Cross-lingual Transfer** In Section 5, we found that even when only Python (40K samples) was used during the fine-tuning phase, the model demonstrated significant improvements in other languages (average score of all languages). We attribute this partly to the rich multilingual corpus embedded in the base model, and partly to the cross-lingual transfer of programming capabilities.
>
> 2. **Programming Grammar**  In Appendix A.10, we conducted an initial analysis of the semantic similarities among different languages in McEval through embeddings. As seen in Figure 14, languages with similar syntax or programming paradigms exhibit higher similarity. We selected training data from several languages in McEval-instruct, which exhibit significant grammatical differences (approximately 10K samples of Python and 1K samples for other languages) and fine-tuned the model. The results are as follows:
>
> | Setting | Python | Scheme | TypeScript | JavaScript | AWK |
> | - | - | - | - | - | - |
> | CodeQwen1.5-base | 38.0 | 34.0 | **52.0** | **52.0** | 38.0 |
> | Only Python | **48.0** | 12.0 | 0.0 | 0.0 | **40.0** |
> | Python+Scheme+TypeScript+AWK | 44.0 | **38.0**| 50.0 | 48.0 | **42.0** |
>
> (1) **Only Python**: When trained using only Python data, the performance on Python and AWK improved. However, this led to the scores for TypeScript and JavaScript dropping to 0. Upon inspection, we found that the generated code for these two languages contained syntax errors (Less data may lead to instability in model training).
> (2) **Python + Other Languages**: When training on a mixture of several languages, Python performance decreased slightly compared to using only Python data, while Scheme performance improved significantly. Furthermore, the syntax generation for TypeScript and JavaScript returned to normal (even without adding JavaScript data, as TypeScript and JavaScript share similar syntax). However, there was no significant improvement compared to the base model.
>
> Thus, fine-tuning multilingual code models presents significant challenges. **Similar languages can provide mutual benefits, while languages with greater differences may negatively impact performance.**
>
> We believe that balancing generation capabilities across different languages still requires significant exploration.
>
> Thank you once again for your detailed and insightful suggestions on our paper. We will integrate the provided feedback into our manuscript to enhance and refine our work further.

---

> > ### Comment · Reviewer_3QMs · 2024-11-25
> > **Respond to Author**
> >
> > Interesting findings! Please update the results in next revision. I'll raise my score to 6.

---

> > > ### Author Response · Authors · 2024-11-27
> > >
> > > Thank you once again for your constructive questions and suggestions on our paper! They have been immensely helpful in improving our work. We have updated the relevant experimental content and included it in Appendix A.10.

---

### Official Review · Reviewer_kkUn · 2024-10-27

**Soundness:** 2
**Presentation:** 3
**Contribution:** 3
**Rating:** 6
**Confidence:** 5

**Summary:**

This paper introduces MCEVAL, a massively multilingual code evaluation benchmark covering 40 programming languages with 16,000 test samples. It aims to advance the evaluation of code LLMs in multilingual scenarios. The benchmark includes tasks for code generation, completion, and explanation. Additionally, the authors present MCODER, a multilingual coder trained on the MCEVAL-INSTRUCT corpora, demonstrating its effectiveness in multilingual programming language generation. In its experiments, this paper highlights the gap between open-source and closed-source models and provides a comprehensive framework for evaluating and improving code LLMs across diverse languages.

**Strengths:**

1. This is truly a BIG project. The authors have made significant contributions to the field of multilingual code generation.

**Weaknesses:**

1. Although this work makes a significant contribution to benchmarking the multilingual code generation capabilities of LLMs, its main contribution is labor-intensive, with limited technical contributions or insights (MCEVAL-INSTRUCT looks similar to MagiCoder by generating instruction data from code snippets).
2. Is it meaningful to benchmark code LLMs on 40 languages? Can the author elaborate on why it is important to measure the generation capabilities of code LLMs in 40 languages simultaneously?
3. Based on the experimental results, MCEVAL and MultiPL-E have similar measurement capabilities, which diminishes the significance of MCEVAL compared to MultiPL-E.
4. It is said that "three volunteers are employed to evaluate the correctness of the benchmark (> 90% accuracy) and correct the errors" but no details are provided. Can the author explain how to ensure that the entire benchmark has a 90% accuracy rate? How was the "90%" quantified (we need ground truth to calculate the 90% right)?

**Questions:**

1. Table 2 shows that MCEVAL has 16,031 questions, but in reality, there are only 2,007 completely independent programming problems. I think that comparing the benchmarks based on independent programming problems would better illustrate the quality of each benchmark, as a benchmark can increase its number of questions by "creating new tasks", but it cannot do the same to generate new independent programming problems.
2. The fonts in the leaderboard tables are too small.
3. The programming tasks differ between languages, making it hard to directly compare a model's capabilities across different languages.

---

> ### Author Response · Authors · 2024-11-18
> **Authors' Response (1/2)**
>
> We greatly appreciate your insightful feedback. Your suggestions have prompted us to reconsider certain aspects of our work. Below, we offer clarifications that we hope will satisfactorily address the concerns raised.
>
> > **Q1**: Its main contribution is labor-intensive, with limited technical contributions or insights
>
> We fully appreciate your concerns.
> 1. Our work is dedicated to building a large-scale multilingual code evaluation dataset to further advance the community’s exploration of multilingual code tasks. As you noted, we have invested substantial effort to ensure the high quality of dataset, including data annotation, testing sandboxes contruction, and code snippet extraction method across 40 languages.
>
> 2. Magicoder is indeed an impressive work with a scientifically efficient approach to code synthesis. Differently, since the created instruction samples across different programming languages emphasize various aspects of coding, we employ cross-lingual code transfer to reduce the gap among multiple languages(Described in Section3.1 and figure 4).
>
> 3. In addition to constructing the datasets (McEval and McEval-Instruct), we conducted several insightful experiments. In Section 5, we explore model performance imbalances, challenges with cross-lingual transfer, and in Appendix A.10, we analyze syntax similarity. We believe these explorations contribute to a more robust multilingual code model by considering cross-lingual transfer, syntax-based data balancing, and effective capability transfer in multilingual training.
>
>
> > **Q2**: Is it meaningful to benchmark code LLMs on 40 languages?
>
> Thank you for the excellent question.
> 1. Today, code LLMs have become highly practical tools for code development. We expect the powerful assistance capabilities of code models not to be limited to popular languages but to extend further, aiding more industries, applications, and developers.
>
> 2. Many current code models claim to support over 80 languages. However, they lack effective evaluations to validate this support. We hope our work can drive advancements in multilingual evaluation.
>
> 3. The development of LLM-based OS automation, AI code editors, and other intelligent agents requires robust multilingual programming capabilities. Some projects or actions demand the collaborative debugging and development of multiple programming languages (e.g. web development).
>
> Therefore, **massive code evaluation** is essential for building **powerful and comprehensive code models capable of handling complex and challenging tasks**. Additionally, our dataset has been used by Qwen-Coder 2.5 and OpenCoder to evaluate the multilingual performance of code models.
>
>
> > **Q3**: MCEVAL and MultiPL-E have similar measurement capabilities
>
> Sorry for any confusion. Regarding the concern you raised, our explanation is as follows:
>
> 1. *MultiPL-E* is indeed a meaningful and solid work. However, MultiPL-E primarily translates from *HumanEval* and *MBPP*, focusing on comparing performance differences between languages. To further enrich the multilingual code evaluation landscape, we have constructed the McEval dataset with questions written by human, offering a broader range of problem types and language diversity (including markup languages, editor languages, and scripting languages).
> Our dataset places a stronger emphasis on evaluating the multilingual programming capabilities of models.
>
> 2. Regarding your mention of *similar measurement capabilities*, as shown in Figure 7(1) of our paper, many models exhibit significant performance differences between MultiPL-E and McEval in Python, likely due to the relative simplicity of the questions in MultiPL-E. Therefore, our data can serve as a valuable supplement.
> Moreover, our leaderboard could reveal some programming language deficiencies in some models. For instance, CodeQwen-1.5 demonstrates outstanding performance on MultiPL-E but performs very poorly in languages not included in MultiPL-E, such as TCL, Elisp, and VimL in McEval. Our proposed McEval can comprehensively evaluate the multilingual capability of the LLM.
>
> 3. Evaluation of three tasks:  In addition to the generation task similar to MultiPL-E, we also provide code explanation and code completion tasks.
>
> We hope that our work, together with outstanding benchmarks like MultiPL-E, will help drive advancements in multilingual programming.

---

> > ### Author Response · Authors · 2024-11-30
> >
> > Hi, Reviewer kkUn,
> >
> > We believe we have addressed your concerns carefully. If you have other questions or comments, please let us know. We are very glad to solve your concerns.
> >
> > Thanks for your insightful suggestions again!

---

> ### Author Response · Authors · 2024-11-18
> **Authors' Response (2/2)**
>
> > **Q4**: How to ensure that the entire benchmark has a 90% accuracy rate?
>
> This is a critical question, and we apologize for not providing sufficient detail in the main text.
>
> The 90% accuracy refers to ensuring that the annotated code passes the corresponding unit tests with a success rate of over 90% before the final review by the three volunteers(senior annotators).
>
> As we mentioned in Appendix A.2.3, we adopt a dual-pass system to control the quality of our benchmark. First, one annotator labels the code and its corresponding unit tests. Then, another annotator independently reviews the annotated code, verifying the annotations for issues, code accuracy, and unit tests. Finally, three senior annotators assess the overall unit test pass rate for the data labeled by the two annotators. If the pass rate exceeds 90%, the senior annotators proceed to further examine and correct any errors. Otherwise, the data is returned to the two annotators for refinement. **Finally, the canonical solution (labeled by annotator) passes all corresponding test cases to ensure the correctness of each created problem (100% pass rate).**
>
>
> > **Q5**: Only 2,007 completely independent programming problems
>
> Thank you for your valuable suggestions!
> We have updated the comparison tables with other works, providing a more detailed breakdown of the number of tasks in each category of our dataset to enable a more objective and comprehensive comparison.
>
> > **Q6**: The fonts in the leaderboard tables are too small
>
> Thank you once again for your excellent advice.
>
> Our original intention was to provide readers with detailed and reliable experimental results, highlighting the highest scores in bold and adding cell comparison colors. However, we realized that too much text led to a reduction in font size.
>
> To make it easier for readers to view and compare, we have redrawn the tables as bar charts. Due to space constraints, we have temporarily placed this content in Appendix A.7 (Figures 14, 15, 16, 17, and 18 on pages 22 to 24 ). We will continue to adjust the placement of the figures in the main text to provide readers with clearer results of the metrics.
>
> > **Q7**: The programming tasks differ between languages
>
> This is an exceptionally valuable question.
>
> Unlike MultiPL-E, we did not emphasize annotating parallel questions between different languages.
> On the one hand, due to the various language-specific features, it is difficult to achieve complete parallelism (e.g., markup languages, scripting languages, editor languages, functional programming). On the other hand, during annotation, we focused more on making the tasks align with real-world language application scenarios (e.g., using awk for text processing, markup languages as carriers of information, and script languages for shell operations), where we aim at creating a language-specific benchmark.
> However, we maintained a similar difficulty ratio for each language, aligning with the overall difficulty ratio (3:1:1) shown in Table 1. For each language, we also preserved an approximate 3:1:1 ratio.
> Therefore, we believe that MultiPL-E is more suitable for comparing model performance across different languages, while our McEval focuses on evaluating the model's overall multilingual programming capability.
>
>
> We appreciate your numerous suggestions for improvement and are actively working to incorporate your comments into an updated draft. The suggestions in Q4, Q5, and Q6 have been updated in the draft. Please kindly review them.

---

> ### Author Response · Authors · 2024-11-30
> **Authors' Response 3**
>
> > **Supplementary response to question Q5**
>
> Thank you again for your valuable suggestions!
> We have reorganized Table 2 and updated some of the data to more objectively compare our work with others in terms of task types, data sources, and the number of problems for each task.
>
> | Benchmark | Task | languages | Data Source |#Questions |
> |-|-|-|-|-|
> |MultiPL-E | Generation | 22 | translate from HumanEval, MBPP | 12,145 |
> |MBXP | Genration, Translation, Robustness, Completion, Summarization | 10 | translate from MBPP | 12,425 (for Generation, other tasks no release.) |
> |HumanEval-X| Generation, Translation | 5 | Hand-Written | 1640(820/820) |
> |HumanEval-XL| Generation | 12 | Hand-Written | 22,080(960 programs with 23 different natural language questions.) |
> | McEval | Genration, Completion, Explanation| 40 | Hand-Written | 16,031 (2007/2007/12017) |
>
> We will update this table in the next version of the paper.

---

> ### Author Response · Authors · 2024-12-02
>
> Hello, Reviewer kkUn,
>
> Thanks again for your insightful comments. As the discussion deadline is coming, please let us know if our responses have addressed your concerns well.

---

> > ### Comment · Reviewer_kkUn · 2024-12-02
> >
> > Thanks for the authors' rebuttal! I'm satisfied with the responses and decided to raise my score to 6.

---

> > > ### Author Response · Authors · 2024-12-02
> > >
> > > Once again, thank you for the detailed and constructive feedback you provided on our work! Your suggestions have greatly contributed to improving our paper!

---

### Official Review · Reviewer_1Wxp · 2024-11-02

**Soundness:** 3
**Presentation:** 4
**Contribution:** 3
**Rating:** 6
**Confidence:** 4

**Summary:**

The MCEVAL benchmark introduces a challenging, multilingual evaluation framework for code LLMs, covering 40 programming languages with 16K test samples. Alongside this, MCEVAL-INSTRUCT - a curated multilingual instruction corpora, and MCODER—a multilingual coder trained on MCEVAL-INSTRUCT are introduced.

**Strengths:**

(1) Extensive study and effort has been spent in generating and documenting this code benchmark.

(2) While HumanEval and MBPP are the popular code programming benchmarks today, Code LM community needs additional multilingual programming benchmark. Hence, this paper addresses a problem statement that is of demand today.

**Weaknesses:**

GPT models, like, gpt-4-1106-preview, has been used to generate the problem description for code instruction corpora. That could have influenced GPT models to lead in the benchmarks with significant performance margins over other models.

**Questions:**

Is there any study done to validate the accuracy or correctness of  MCEVAL benchmarks?

---

> ### Author Response · Authors · 2024-11-18
> **Authors' Response**
>
> Thank you very much for your positive evaluation of our work! Below, we provide responses to your questions and hope they will help address any concerns you may have.
>
> > **Q1**: Influenced GPT models to lead in the benchmarks with significant performance margins over other models.
>
> This is an extremely insightful question, and we provide more detail in the Appendix.
>
> We describe the annotation process in the section Appendix A.2.2,
> When annotating the test dataset McEval, the annotators drew inspiration from the reference website and manually created the questions with the assistance of GPT-4-Turbo. During the annotation process, we observed that many questions still contained numerous errors, even when generated by GPT-4, requiring manual corrections by the annotators.
>
> Furthermore, as shown in the below table, other models (such as Claude, OpenAI-O1, DeepSeek-Coder, etc., which were not used during the annotation process) also demonstrated exceptional performance on the benchmark.
>
> | Model | McEval (AVG, Pass@1) |
> | - | - |
> | gpt-4-turbo-231106         | 63.4 |
> | gpt-4o-2024-08-06          | 65.8 |
> | gpt-4o-mini-2024-07-18     | 58.1 |
> | o1-mini                    | 58.1 |
> | o1-preview                 | 71.1 |
> | claude-3-5-sonnet-20240620 | 67.3 |
> | claude-3-5-sonnet-20241022 | 66.5 |
> | DS-Coder-V2-Instruct       | 62.9 |
>
> Your insights are highly valuable. We acknowledge that we cannot overlook this aspect, **we will also prepare the instruction corpora created by other closed-source LLMs (e.g. Claude) and open-source models (e.g. DeepSeek-Coder) for diversity to avoid the unbalanced influence from the GPT-4.**
>
> We greatly appreciate your suggestion. We will explore the potential impact of model-generated data on evaluation fairness in future research.
>
> > **Q2**: Is there any study done to validate the accuracy or correctness of MCEVAL benchmarks?
>
> We fully appreciate your concerns.
>
> Our quality control process for the dataset is primarily focused on the annotation stage. As detailed in Appendix A.2.3, we employ a dual-pass annotation system to ensure the reliability of our benchmark. Initially, one annotator labels the code along with its corresponding unit tests. This is followed by an independent review by a second annotator, who checks the annotations for potential issues, code accuracy, and the validity of the unit tests. Subsequently, a team of three senior annotators evaluates the overall unit test pass rate for the data produced by the two annotators. If the pass rate exceeds 90%, the senior annotators conduct further reviews and corrections. Otherwise, the data is sent back to the two annotators for refinement and re-evaluation. **Finally, the canonical solution (labeled by annotator) passes all corresponding test cases to ensure the correctness of each created problem (100% pass rate).**

---

> > ### Comment · Reviewer_1Wxp · 2024-11-26
> >
> > Thanks for clarifying.

---

### Official Review · Reviewer_npVu · 2024-11-04

**Soundness:** 4
**Presentation:** 4
**Contribution:** 4
**Rating:** 8
**Confidence:** 5

**Summary:**

The authors propose a new code benchmark called McEval. The proposed benchmark has 40 programming languages with 16k samples. The benchmark spans code generation, code completion and code understanding tasks. The authors also introduce mCoder trained on the McEval-Instruct corpora.

**Strengths:**

The paper is well-written and is very easy to read and follow. The paper is also a significant leap towards better evaluation for code models. Current models are mostly evaluated on smaller benchmarks like HumanEvalPack or MBPP which have very few languages (<10) and this limits the ability to effectively evaluate code models.

The authors' work is quite significant since it introduces a new benchmark with 40 programming languages which is more than 4x of any previous benchmarks. The paper also presents an instruction corpora.

I overall like the paper and the amount of experiments done by the authors.

**Weaknesses:**

The instruction corpora is generated using GPT-4 and thus is commercially unusable. I understand that collecting good human annotated instructions is a significant undertaking and might be infeasible.

**Questions:**

1. Is the dataset being released?
2. Have you evaluated [Granite code models](https://huggingface.co/collections/ibm-granite/granite-code-models-6624c5cec322e4c148c8b330), it would be nice to have them in the report since there are very few SOTA code models available under Apache 2.0 license?
3. Are the human annotators fairly compensated?

---

> ### Author Response · Authors · 2024-11-18
> **Authors' Response**
>
> Thank you very much for your high recognition of our work! Below are the answers to your questions, and we hope they will clarify any doubts you may have.
>
> > **Q1**: The instruction corpora is commercially unusable.
>
> We completely understand your concerns.
> However, so far, the most efficient way to generate synthetic data remains the use of powerful models (e.g. GPT-4). With the release of large open-source models such as Llama-3.1-405B and DeepSeek-238B, we believe that more efficient, commercially viable methods for generating high-quality synthetic data will emerge in the future (We will also provide the instruction corpora using these open-source models such as DeepSeek-238B for commercial use). Additionally, we consider leveraging large models as teachers to train smaller models for data synthesis to be a highly promising direction for exploration!
>
> > **Q2**: Is the dataset being released?
>
> Yes, we have open-sourced **McEval**, **McEval-Instruct** (Huggingface), the sandbox environment image for evaluation (Docker), the code for training **mCoder** (Github), code snippet extraction, and evaluation scripts (Github). Additionally, we also have maintained a leaderboard in the Internet.
>
> > **Q3**: Have you evaluated Granite code models(available under Apache 2.0 license).
>
> Thank you for your valuable suggestions!
>
> We have been actively engaging in testing more code models. We have tested the Granite series code models on code generation task as below and updated the results in the paper (Table 3), which demonstrated excellent performance. We will also add the performance of the other two tasks to the paper later.
>
> | Model | McEval (AVG, Pass@1)|
> | - | - |
> | o1-preview                 | 71.1 |
> | claude-3-5-sonnet-20240620 | 67.3 |
> | gpt-4o-2024-08-06          | 65.8 |
> | claude-3-5-sonnet-20241022 | 66.5 |
> | DeepSeek-Coder-V2-Instruct-238B  | 62.9 |
> | gpt-4o-mini-2024-07-18     | 58.1 |
> | o1-mini                    | 58.1 |
> | **Granite-34B-code-instruct-8K**  | 42.2 |
> | **Granite-20B-code-instruct-8K**  | 38.3 |
> | **Granite-8B-code-instruct-4K**   | 36.5 |
> | **Granite-3B-code-instruct-128K** | 28.0 |
> | CodeGemma-7B-Instruct | 30.7 |
> | CodeLlama-13B-Instruct | 27.7 |
> | CodeLlama-7B-Instruct | 24.6 |
>
> During this period, we also supplemented the study with newer models, such as GPT-o1, Claude-sonnet. We hope that our work can contribute to the research on multilingual code understanding, we also hope that more models like Granite, which feature excellent open-source licenses, will contribute to research in the code community!
>
> > **Q4**: Are the human annotators fairly compensated?
>
> This is a very important ethical issue!
> Yes, we discussed reasonable annotation compensation with the annotators before the annotation work began. We describe the compensation in the section Appendix A.2.4 ANNOTATION COSTS. Specifically, we paid all the annotators the equivalent of $6 per question and provided them with a comfortable working environment, free meals, and souvenirs.
>
> Once again, we sincerely thank you for your high praise of our work. We will actively consider incorporating your valuable feedback into our paper and continue testing more excellent open-source models. We hope to make further contributions to multilingual research in the field of code.

---

> > ### Comment · Reviewer_npVu · 2024-11-23
> >
> > Thanks, I am quite satisfied with the responses.

---

### Author Response · Authors · 2024-11-18
**General Response by Authors**

We sincerely thank all reviewers for their time and constructive feedback. In response to these valuable suggestions, we have made the following revisions to the paper, highlighted in yellow for clarity. We list these changes below.
1. We updated Table 2 (Page 3) by adding the number of data for each task to provide a more objective comparison, as suggested by Reviewer kkUn.
2. We tested the Granite Code Models[1] (ranging from 3B to 34B) and added the results to Table 3 (Page 6), as suggested by Reviewer npVu.
3. As suggested by Reviewers 1Wxp and kkUn, we revised Appendix A.2.3 QUALITY CONTROL (Page 18), providing a clearer description to address concerns raised by reviewers about the data annotation process.
4. We redraw some model results as bar charts (Figures 14, 15, 16, 17, and 18, Pages 22–24), which are temporarily placed in the appendix. This aims to address Reviewer kkUn’s concerns about the small font size in the experiment result tables and provide a more intuitive comparison for readers. Due to time constraints, we will continue to adjust the number of models in the figures and their placement in the main text.
5. As noted by Reviewer kkUn and 3QMs, our synthetic data generation method shares similarities with Magicoder[2] and the concurrent work BigCodeBench[3]. We have added a description of Magicoder and BigCodeBench in the "Detailed Related Work" section of the Appendix (Page 26) and summarized the key differences: our approach extends to a broader range of programming languages.

We sincerely appreciate the positive feedback on McEval and are dedicated to further advancing research in multilingual programming. We remain committed to continuously maintaining and improving the benchmark to support ongoing research and development.

**Reference**

[1] Mishra M.,et al. "Granite code models: A family of open foundation models for code intelligence". arXiv 2024.

[2] Wei Y.,et al. "Magicoder: Empowering code generation with oss-instruct". ICML 2024.

[3] Zhuo T Y.,et al. "Bigcodebench: Benchmarking code generation with diverse function calls and complex instructions". arXiv 2024.

---

### Meta-Review · Area_Chair_mZJV · 2024-12-22

**Metareview:**

> The authors propose a new code benchmark called McEval. The proposed benchmark has 40 programming languages with 16k samples. The benchmark spans code generation, code completion and code understanding tasks. The authors also introduce mCoder trained on the McEval-Instruct corpora.

The reviewers all accept the paper. Code generation benchmark are severely underserved beyond Python (and a few other popular languages). This benchmark seems to be of relevant scope and size to help the whole field of code generation measure progress.

**Additional Comments On Reviewer Discussion:**

Reviewers kkUn, npVu and 1Wxp all took part in the rebuttal discussion.

---

### Decision · Program_Chairs · 2025-01-22

Accept (Poster)